# Experimental evidence for cancer resistance in a bat species

Rong Hua[1,2,4], Yuan-Shuo Ma[1,2,4], Lu Yang[1,2,4], Jun-Jun Hao[1], Qin-Yang Hua[1,2], Lu-Ye Shi[1], Xiao-Qing Yao[1,2], Hao-Yu Zhi[1,2] & Zhen Liu ⓘ[1,3] ✉

Mammals exhibit different rates of cancer, with long-lived species generally showing greater resistance. Although bats have been suggested to be resistant to cancer due to their longevity, this has yet to be systematically examined. Here, we investigate cancer resistance across seven bat species by activating oncogenic genes in their primary cells. Both in vitro and in vivo experiments suggest that *Myotis pilosus* (MPI) is particularly resistant to cancer. The transcriptomic and functional analyses reveal that the downregulation of three genes (*HIF1A*, *COPS5*, and *RPS3*) largely contributes to cancer resistance in MPI. Further, we identify the loss of a potential enhancer containing the HIF1A binding site upstream of *COPS5* in MPI, resulting in the downregulation of *COPS5*. These findings not only provide direct experimental evidence for cancer resistance in a bat species but also offer insights into the natural mechanisms of cancer resistance in mammals.

Bats comprise ~20% (>1400 species) of all living mammalian species and are considered one of the most well-adapted mammalian groups in their environments[1]. Their success can be attributed to several remarkable adaptations, such as true self-powered flight, echolocation, virus tolerance, and extraordinary longevity[1]. Most bat species have lifespans that are more than three times longer than those of similar-sized mammals[2,3]. In particular, 18 of 19 mammalian species with lifespans longer than humans relative to their body size are bat species[4].

It is worth noting that long-lived mammalian species, such as naked mole rats and blind mole rats, often appear to be remarkably cancer-resistant due to their evolved longevity mechanisms that are closely linked to cancer resistance, including repressed telomerase activity, inhibited cell proliferation, and enhanced genome stability[5,6]. This suggests that bats may have also evolved cancer resistance, particularly for the long-lived species[5,6]. Molecular and genomic studies provide accumulating evidence in support of this hypothesis. For instance, oncogenic microRNAs are downregulated, while cancer-resistant microRNAs are upregulated in the long-lived bat species *Myotis myotis*[7]. Additionally, adaptive changes have been identified in the growth hormone receptor and P53 in the long-lived bat species,

both of which are closely associated with cancer resistance[8,9]. Nevertheless, there is limited knowledge regarding experimental evidence for cancer resistance in bats, which necessitates systematic verification because the discovery of mammalian species with anticancer properties could provide valuable insights into the natural mechanisms of cancer resistance.

In this study, we conduct in vitro and in vivo experiments on seven bat species to investigate their cancer resistance. We find that the big-footed bat, *Myotis pilosus*, exhibits particularly high resistance to cancer. Our findings indicate that the anticancer mechanisms in the big-footed bat involve the downregulation of three genes associated with cancer: *HIF1A*, *COPS5*, and *RPS3*. Specifically, the reduced expression of *COPS5* in the big-footed bat is most likely due to the loss of a potential enhancer that contains the binding site for the transcription factor HIF1A.

## Results

### Fibroblasts of the big-footed bat (*Myotis pilosus*) exhibit resistance to malignant transformation

To assess the potential of bat species to be resistant to cancer, we conducted a systematic investigation using primary skin fibroblasts

[1]State Key Laboratory of Genetic Resources and Evolution, Kunming Institute of Zoology, Chinese Academy of Sciences, Kunming, China. [2]Kunming College of Life Science, University of Chinese Academy of Sciences, Beijing, China. [3]Yunnan Key Laboratory of Biodiversity Information, Kunming, China. [4]These authors contributed equally: Rong Hua, Yuan-Shuo Ma, Lu Yang. ✉e-mail: zhenliu@mail.kiz.ac.cn

from seven bat species, including the big-footed bat (*Myotis pilosus*; MPI for short), the Szechwan myotis (*Myotis altarium*; MAL), the least horseshoe bat (*Rhinolophus pusillus*; RPU), the greater horseshoe bat (*Rhinolophus ferrumequinum*; RFE), the Chinese rufous horseshoe bat (*Rhinolophus sinicus*; RSI), the great leaf-nosed bat (*Hipposideros armiger*; HAR), and the Leschenault's Rousette (*Rousettus leschenaultii*; RLE). We also used the laboratory mouse (*Mus musculus*) as a control. Although most cancers in humans originate from epithelial cells and fibroblasts are part of the tumor microenvironment, studying fibroblasts can provide insights into whether a species tends to have anticancer properties by assessing their resistance to malignant transformation. For example, previous studies have shown that the expression of oncogenic HRAS(G12V) and SV40 large antigen (SV40 LT) effectively induces malignant transformation in mouse fibroblasts[10,11]. However, this transformation is not observed in fibroblasts derived from naked mole-rats and blind mole-rats, which are well-known for their anticancer characteristics[12,13]. Therefore, we generated stable cell lines of fibroblasts that continuously express the oncogenic HRAS(G12V) and SV40 LT to determine if this combination would enable anchorage-independent growth to bat fibroblasts. To rule out the effects of varying expression levels[14], we conducted immunoprecipitation to quantify the protein levels of both genes (Fig. 1a). Our results indicated successful expression of HRAS(G12V) and SV40 LT at similar levels in the stable cell lines from all eight mammalian species (Supplementary Fig. 1). The subsequent anchorage-independent growth assay showed that fibroblasts from mouse and six bat species (MAL, RPU, RFE, RSI, HAR, and RLE) formed large colonies in soft agar compared to fibroblasts without HRAS(G12V) and SV40 LT (Fig. 1b). However, MPI fibroblasts expressing HRAS(G12V) and SV40 LT formed significantly smaller colonies than those from mouse and other six bat species (*P* < 0.001; two-tailed Student's *t* tests; Fig. 1b, c). These findings were further confirmed by establishing primary fibroblasts from other tissues of MPI, such as the

intestine and tail, and repeating the experiments. Consistently, these fibroblasts from MPI failed to form large colonies (Supplementary Fig. 2).

To further validate the resistance of MPI cells to malignant transformation, we introduced luciferase into mouse (MSF^HRAS\SV40LT) and MPI (MPI-SF^HRAS\SV40LT) fibroblasts expressing HRAS(G12V) and SV40 LT, as the intensity of the luciferase fluorescence can be used to reflect the size of the xenograft or tumor. Subsequently, MSF^HRAS\SV40LT and MPI-SF^HRAS\SV40LT were subcutaneously injected into immunodeficient mice to form xenografts, respectively. One week after injection, there was no significant difference in fluorescence intensity between the xenografts derived from MSF^HRAS\SV40LT and MPI-SF^HRAS\SV40LT (Fig. 2a, b). However, after 2 and 3 weeks, tumors derived from MSF^HRAS\SV40LT exhibited rapid growth with a notable increase in fluorescence, while MPI-SF^HRAS\SV40LT formed significantly smaller tumors (*P* < 0.05; two-tailed Student's *t* tests; Fig. 2a, b). Furthermore, the tumors were weighed 3 weeks after cell injection, demonstrating that the tumors derived from MPI-SF^HRAS\SV40LT were indeed significantly smaller and lighter than those derived from MSF^HRAS\SV40LT (*P* = 2.21E-05; two-tailed Student's *t* tests; Fig. 2c, d). These findings reveal that the overexpression of oncogenic HRAS(G12V) and SV40 LT fails to promote the malignant transformation of MPI cells compared to mouse and other six bat species, suggesting that the big-footed bat, *Myotis pilosus* (MPI), has evolved cancer resistance.

## Downregulated expression of *HIF1A*, *COPS5*, and *RPS3* for the resistance of MPI cells to malignant transformation

To investigate the molecular mechanisms underlying the resistance of MPI cells to malignant transformation, we conducted transcriptome sequencing of fibroblasts from the eight species mentioned above, with three biological replicates. Principal component analysis revealed that the three biological replicates from the same species tended to cluster together, indicating the reliability and repeatability of our

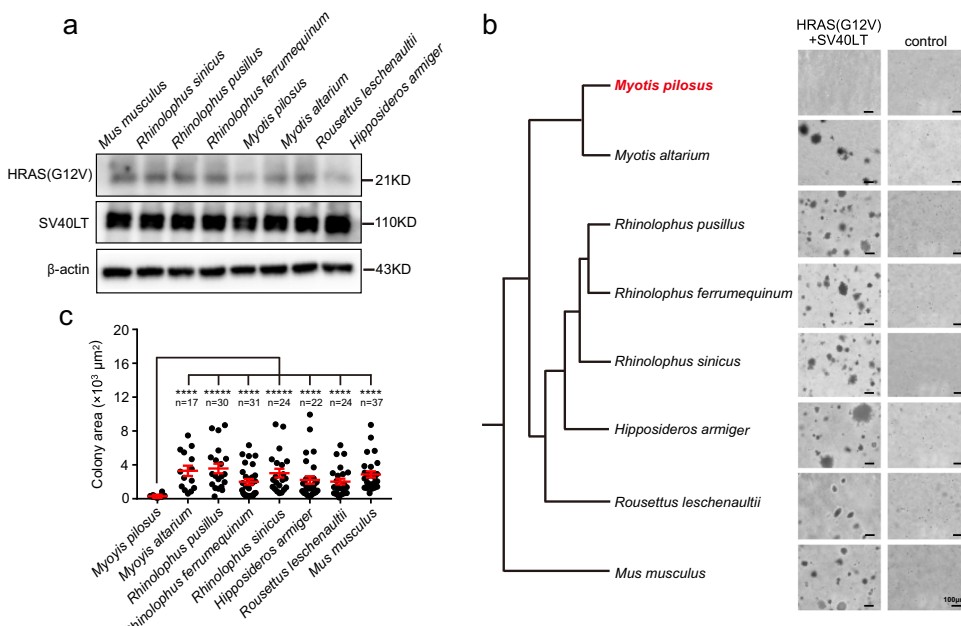

**Fig. 1 | Fibroblasts of the big-footed bat (*Myotis pilosus*; MPI for short) are resistant to malignant transformation. a** Immunoblotting shows no significant differences in the expression of oncogenic HRAS(G12V) and SV40 LT among fibroblasts from seven bat species, as well as mouse. The experiment was repeated independently at least three times with similar results. **b** Anchorage-independent growth assay of the fibroblasts with and without (control) expressing HRAS(G12V) and SV40 LT for seven bat species and mouse. MPI fibroblasts expressing HRAS(G12V) and SV40LT (MPI-SF^HRAS\SV40LT) form remarkably smaller colonies

compared to the other bat species and the control. The representative microphotographs of the colonies grown in soft agar after 4 weeks are shown at 10× magnification, with a scale bar of 100 μm. **c** Quantitative analysis reveals that the area of the colonies formed by MPI-SF^HRAS\SV40LT is significantly smaller than those from the other species. The *n* values represent the number of colonies from three independent experiments. All data are presented as mean ± SD. The *P* values are from two-tailed Student's *t* tests. ****P < 0.0001. Source data are provided as a Source Data file.

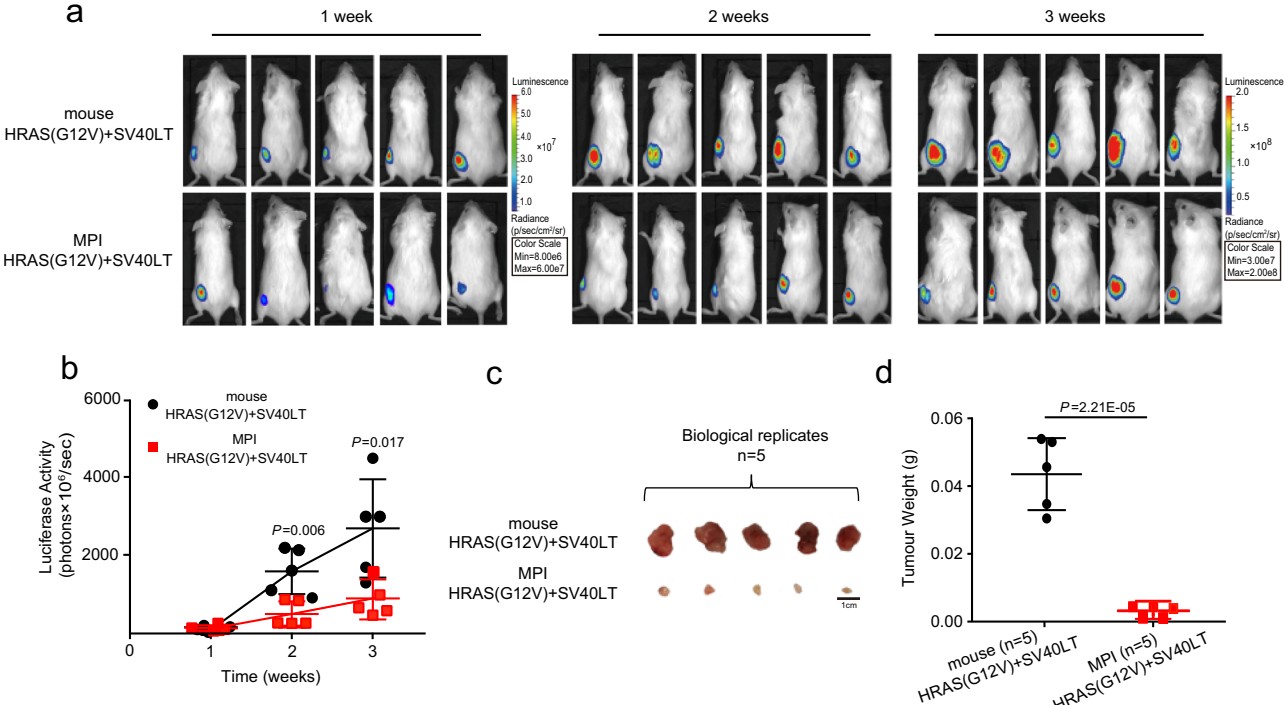

**Fig. 2 | Mouse xenograft assay shows that the MPI fibroblasts expressing oncogenic HRAS(G12V) and SV40 LT are unable to form tumors.**
**a** Representative images of bioluminescence signals in immunodeficient mice with xenografts derived from MSF[HRAS\SV40LT] and MPI-SF[HRAS\SV40LT] are captured at 1, 2, and 3 weeks after subcutaneous implantation. **b** The luciferase activity of the xenografts derived from MSF[HRAS\SV40LT] is significantly higher than that of the xenografts derived from MPI-SF[HRAS\SV40LT] at 2 and 3 weeks after subcutaneous implantation.

**c** Representative images of the xenografts derived from MSF[HRAS\SV40LT] and MPI-SF[HRAS\SV40LT] after 3 weeks of subcutaneous implantation are shown. Scale bar: 1 cm. **d** The xenografts derived from MSF[HRAS\SV40LT] are significantly heavier than those derived from MPI-SF[HRAS\SV40LT] 3 weeks post subcutaneous implantation. All data are presented as mean ± SD. The $P$ values are from two-tailed Student's $t$ tests. The number of dots ($n = 5$) represents the number of biologically independent animals. Source data are provided as a Source Data file.

transcriptomic data (Supplementary Fig. 3). To identify the genes in MPI that exhibited different expression patterns compared to mouse and other bat species, we employed signed weighted gene co-expression network analysis (WGCNA)[15] across the 24 samples. This analysis identified 34 gene co-expression modules, which were numerically labeled by different colors (Fig. 3a; Supplementary Data 1). We then calculated the module eigengene (ME) for each module. The ME represents the first principal component of a module and reflects the overall expression pattern of the genes within the module. Among these modules, M1 and M2 exhibited the highest and lowest ME values on average for MPI, respectively (Fig. 3b). Indeed, the ME value for M1 was significantly higher for MPI than for other species, while the ME value for M2 was significantly lower in MPI compared to for other species ($P < 0.001$; two-tailed Student's $t$ tests; Fig. 3c). To confirm the findings from WGCNA, we examined the differentially expressed genes between these species and observed that 93.8% of genes in the M1, which had the highest ME value, were significantly upregulated in MPI, and 42.9% of genes in the M2, which had the lowest ME value, were significantly downregulated in MPI (Supplementary Data 1). These results were largely aligned with the co-expression trends reflected through the WGCNA analysis.

Further, we constructed a protein-protein interaction network using the STRING database[16] for the genes involved in M1 and M2 (Supplementary Data 2). We then calculated the degree and betweenness centrality for each of the 178 nodes, which are commonly used to identify hub genes in the network[17]. The top 5 hub genes in the network were identified as *HIF1A*, *EP300*, *EIF5B*, *COPS5*, and *RPS3* (Fig. 4a). Notably, all five of these genes were from M2 and were indeed found to be significantly downregulated in MPI compared to mouse and other bat species (Supplementary Fig. 4). To explore the potential

relationship between these genes and tumor development, we collected survival times of cancer patients across 21 tumor types from the KMplot database[18] for the 350 genes involved in M1 and M2 (Supplementary Data 3). We counted the number of tumor types in which the upregulation of a gene in M1 or the downregulation of a gene in M2 significantly prolonged the survival time of cancer patients. The frequency distribution of the ratios of the number of these tumor types to the total number of tumor types with significant survival rates across the 350 genes showed that the lower expression of *HIF1A*, *EP300*, *EIF5B*, *COPS5*, and *RPS3* generally increased the proportion of patients with improved survival (Fig. 4b). In particular, the proportions were significantly higher for *HIF1A* and *EIF5B* compared to other genes, although the difference was only marginally significant ($P = 0.05$; Permutation test; Fig. 4b). To provide additional evidence supporting the influence of gene expression regulation, rather than sequence alterations, on cancer progression, we detected evolutionary pressures on the protein-coding regions of the five candidate genes using PAML[19]. Our analyses revealed that none of these genes were under positive selection or exhibited accelerated evolution in MPI (Supplementary Data 4). These results suggest that the downregulation of these genes may be involved in the resistance of MPI cells to malignant transformation.

To test this hypothesis, we used CRISPR-Cas9 gene-editing technology to inhibit the expression of *HIF1A*, *COPS5*, *RPS3*, *EP300*, and *EIF5B* in MSF[HRAS\SV40LT]. Our results showed that the suppression of *HIF1A*, *COPS5*, and *RPS3* expression significantly inhibited cell proliferation ($P < 0.05$; two-tailed Student's $t$ tests; Fig. 4c). However, the downregulation of *EP300* and *EIF5B* had no remarkable effect on cell proliferation (Supplementary Fig. 5). Notably, these two genes were up-regulated during aging in the long-lived bat (*Myotis myotis*)[20], suggesting their potentially pleiotropic roles in the bat lifespan. To further

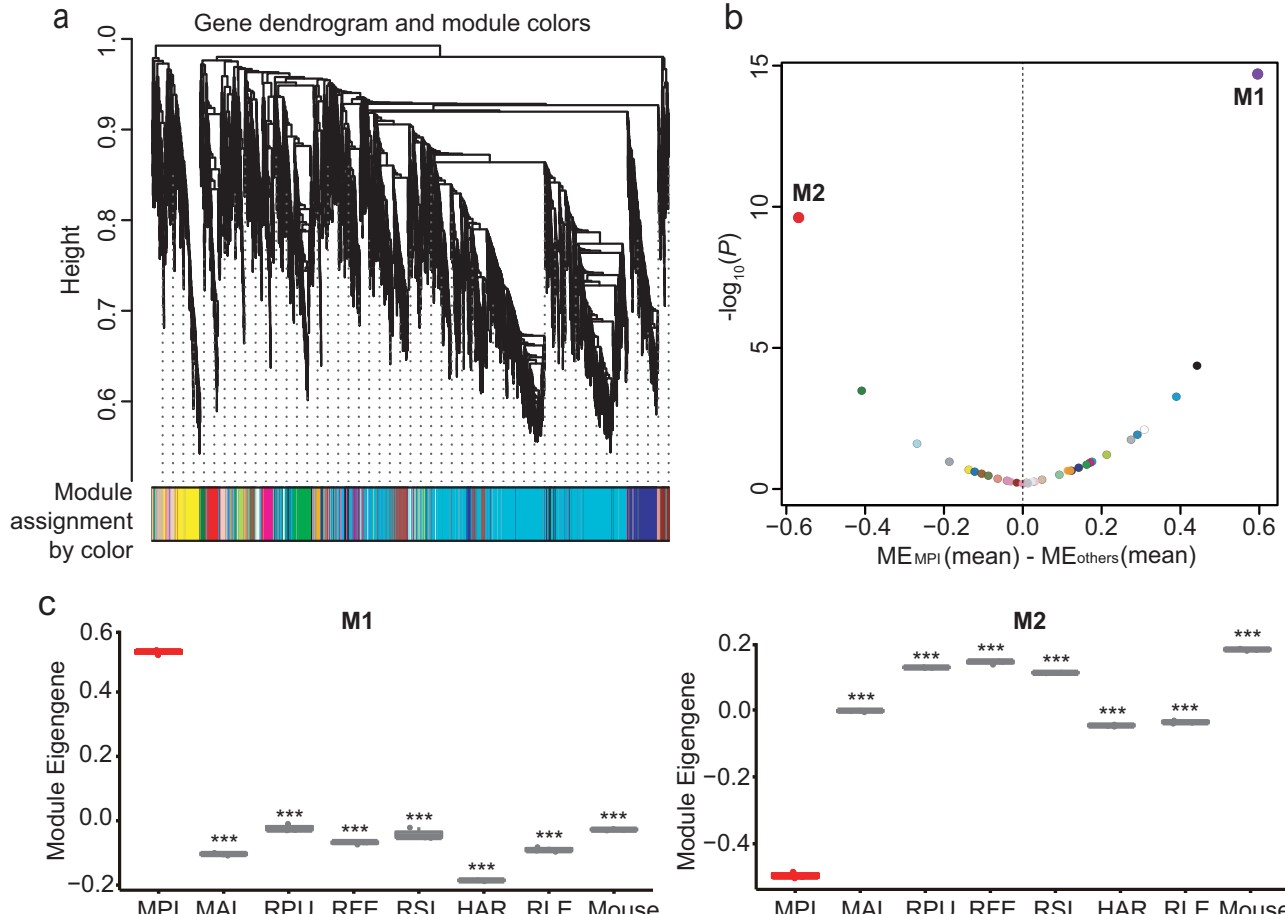

**Fig. 3 | Identification of differential gene co-expression modules between MPI and the other bat species. a** Network analysis dendrogram showing modules based on the co-expression topological overlap of genes among different bat species and mouse. The color bars below give information on module membership. **b** The distribution of mean differences of the module eigengene (ME) between MPI and other species across 34 gene co-expression modules. The upregulated module M1 and downregulated module M2 in MPI are highlighted. **c** The ME value of M1 is significantly larger for MPI than those for other bat species and mouse. The ME

value of M2 is significantly less for MPI than those for other bat species and mouse. The Szechwan myotis (*Myotis altarium*; MAL), the least horseshoe bat (*Rhinolophus pusillus*; RPU), the greater horseshoe bat (*Rhinolophus ferrumequinum*; RFE), the Chinese rufous horseshoe bat (*Rhinolophus sinicus*; RSI), the great leaf-nosed bat (*Hipposideros armiger*; HAR), and the Leschenault's Rousette (*Rousettus leschenaultii*; RLE). The *P* values are from two-tailed Student's *t* tests. \*\*\**P* < 0.001. Source data are provided as a Source Data file.

investigate the involvement of *HIF1A*, *COPS5*, and *RPS3* in the resistance of MPI cells to malignant transformation, we overexpressed *HIF1A*, *COPS5*, and *RPS3* in MPI-SF$^{HRAS\backslash SV40LT}$ (Supplementary Fig. 6a). The overexpression of *HIF1A* and *RPS3* significantly increased cell proliferation (*P* < 0.05; two-tailed Student's *t* tests; Fig. 4d). When the three genes were overexpressed together, the cell proliferative effect was even more pronounced (*P* < 0.01; two-tailed Student's *t* tests; Fig. 4d). Furthermore, the combined overexpression of the three genes in MPI-SF$^{HRAS\backslash SV40LT}$ resulted in significantly larger soft agar colonies and tumor size in mice compared to the control groups (*P* < 0.001; two-tailed Student's *t* tests; Supplementary Fig. 6b, c, d, e, f, and g). Taken together, these results strongly suggest that the downregulation of *HIF1A*, *COPS5*, and *RPS3* is involved in the resistance of MPI cells to malignant transformation.

### Accelerated evolution of an evolutionarily conserved non-coding element is linked to the resistance of MPI cells to malignant transformation

To investigate the molecular basis of the downregulated expression of *HIF1A*, *COPS5*, and *RPS3* in MPI, we focused on evolutionarily conserved non-coding elements (CNEs), as these elements often overlap with cis-

regulatory elements and may function due to their evolutionary sequence conservation driven by purifying selection[21,22]. Since MPI genomic data was unavailable for screening genome-wide CNEs, we sequenced its genome using PacBio technology, generating 206.4 Gb long-read sequences. The assembly was further polished with Illumina short reads and showed relatively high continuity (contig N50: ~80 Mb). Finally, the assembled MPI genome was ~2 Gb in size (Supplementary Data 5). Based on the analysis of the benchmarking of universal single-copy orthologs (BUSCO), we retrieved 96.3% of complete BUSCO genes (Supplementary Data 6). By combining the high-quality MPI genome with the genomic data of six species involved in our experiments, as well as human, we generated a genome alignment (Fig. 5a). Following the pipeline from a previous study[22], we identified a total of 437,414 CNEs among these species and obtained 20,231 CNEs that had undergone accelerated evolution in MPI.

To ascertain the potential regulatory activity of these CNEs, we conducted ATAC-seq (Assay for Transposase-Accessible Chromatin using Sequencing) on MSF and MPI-SF to detect accessible chromatin regions. These regions are often associated with DNA regions that possess gene regulatory activity[23]. We used MACS2[24] to analyze the genomic regions of open chromatin, known as ATAC-seq peaks. To avoid any

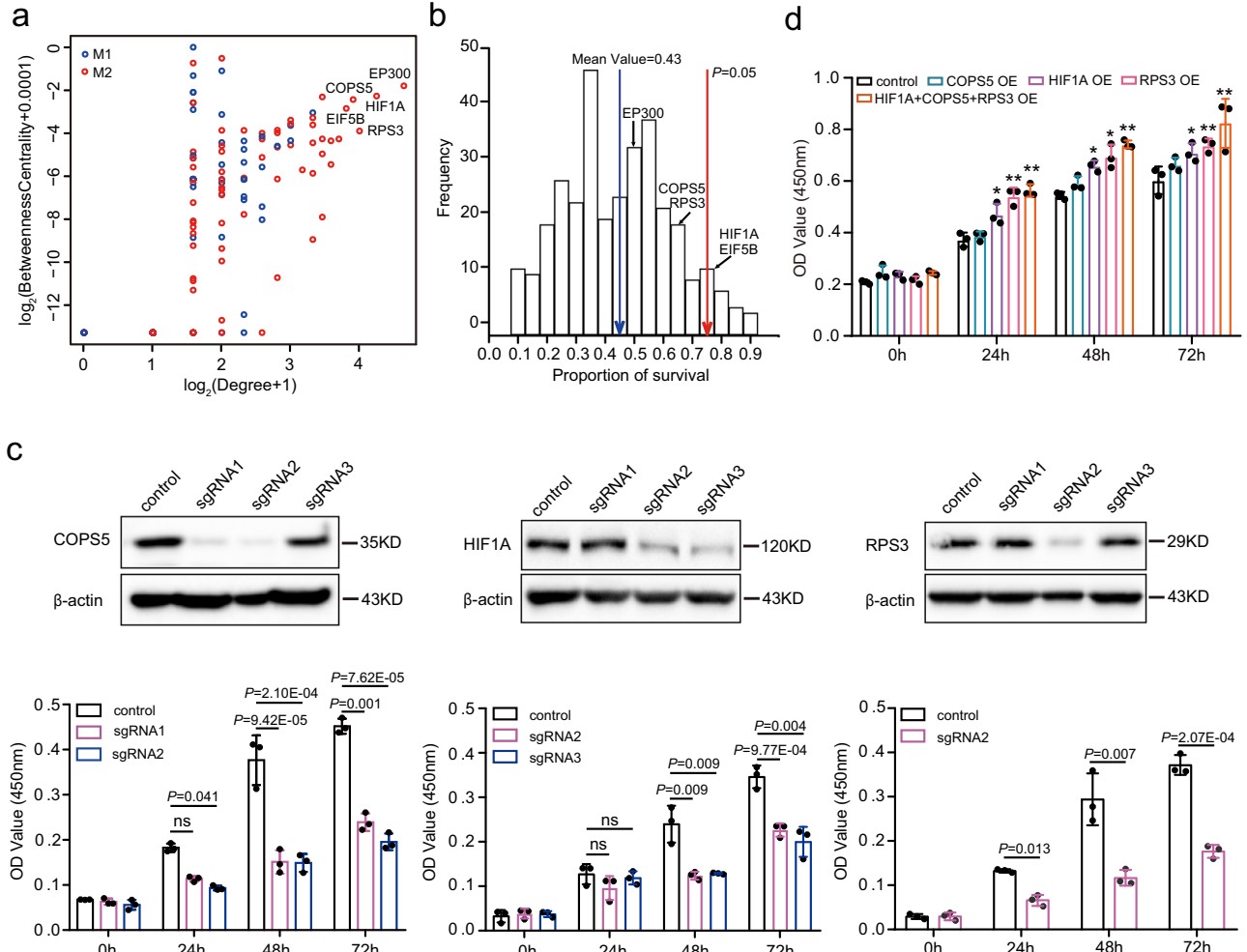

**Fig. 4 | The downregulation of *HIF1A*, *COPS5*, and *RPS3* is associated with the resistance of MPI cells to malignant transformation. a** Protein-protein interaction network analysis for the genes involved in M1 and M2. The correlation between degree and betweenness centrality for each node. The top 5 genes are labeled by names. **b** According to the KMplot database[21], the number of cancer types was counted if the upregulated expression of a gene within M1 or the downregulated expression of a gene within M2 significantly prolonged the survival time of cancer patients. The frequency distribution of the ratios of the number of these tumor types to the total number of tumor types with significant survival rates across 350 genes involved in co-expression modules M1 and M2 is shown. The lower expression of the top 5 hub genes is generally associated with a higher proportion of survivors with tumors. **c** Immunoblotting indicates that sgRNAs effectively suppress the expression of *COPS5*, *HIF1A*, and *RPS3*. The suppression of *COPS5*, *HIF1A*, and *RPS3* expression significantly inhibits cell proliferation. **d** The over-expression of *COPS5*, *HIF1A*, and *RPS3* is linked to cell proliferation. The number of dots (*n* = 3) in (**c**) and (**d**) represents the number of independent experiments. All data are presented as mean ± SD. OE means overexpression. The *P* values are from two-tailed Student's *t* tests. *$P < 0.05$, **$P < 0.01$. Source data are provided as a Source Data file.

potential bias from cross-species comparison, we only focused on genomic regions that lacked peaks in two biological replicates of MPI-SF but had peaks in MSF. This approach resulted in the identification of 106 ATAC-seq peaks within the 300 kb region around *HIF1A*, *COPS5*, and *RPS3*. To further determine their regulatory activity, we overlapped these peaks with candidate cis-regulatory elements (cCREs) derived from the ENCODE database[25]. As a result, we detected 46 putative regulatory regions that contained cCREs (Supplementary Data 7).

Next, we overlaid these putative regulatory regions with the accelerated CNEs in MPI and identified one potential regulatory element, CNE143336, for HIF1A, and one potential regulatory element, CNE563305, for RPS3 (Supplementary Data 7). To assess the regulatory activity of the accelerated CNEs in MPI, we conducted a dual-luciferase assay in both HEK 293T and the fibroblast cell line NIH 3T3. CNE143336 is located -190 kb downstream of *HIF1A* (Fig. 5b). This element and its corresponding sequences from mouse and RSI, a bat species not resistant to malignant transformation, exhibited significantly enhanced activity in both HEK 293T and NIH 3T3 compared to the

negative control ($P < 0.001$; Student's *t* tests; Fig. 5c). Importantly, the enhancer activity of CNE143336 was significantly lower than that of its corresponding sequences from mouse and RSI in both HEK 293T and NIH 3T3 ($P < 0.05$; two-tailed Student's *t* tests; Fig. 5c). To investigate whether the accelerated evolution of CNE143336 inhibits the expression of *HIF1A* in MPI, we utilized CRISPR-dCas9 technology to block its regulatory activity (Fig. 5d). Guide RNAs were designed based on the sequence of CNE143336. The results demonstrated that when the regulatory activity of CNE143336 was blocked, the expression level of *HIF1A* significantly decreased ($P < 0.001$, two-tailed Student's *t* test; Fig. 5e, f). These findings confirm that the accelerated evolution of CNE143336 indeed affects the expression of *HIF1A* in MPI. Unfortunately, we failed to amplify CNE563305 to perform the dual-luciferase assay, likely due to its high GC content (Supplementary Fig. 7).

**Downregulated expression of *COPS5* reduces tumor size**

If the decreased expression of *HIF1A*, *COPS5*, and *RPS3* is linked to the resistance of MPI cells to malignant transformation, it is expected that

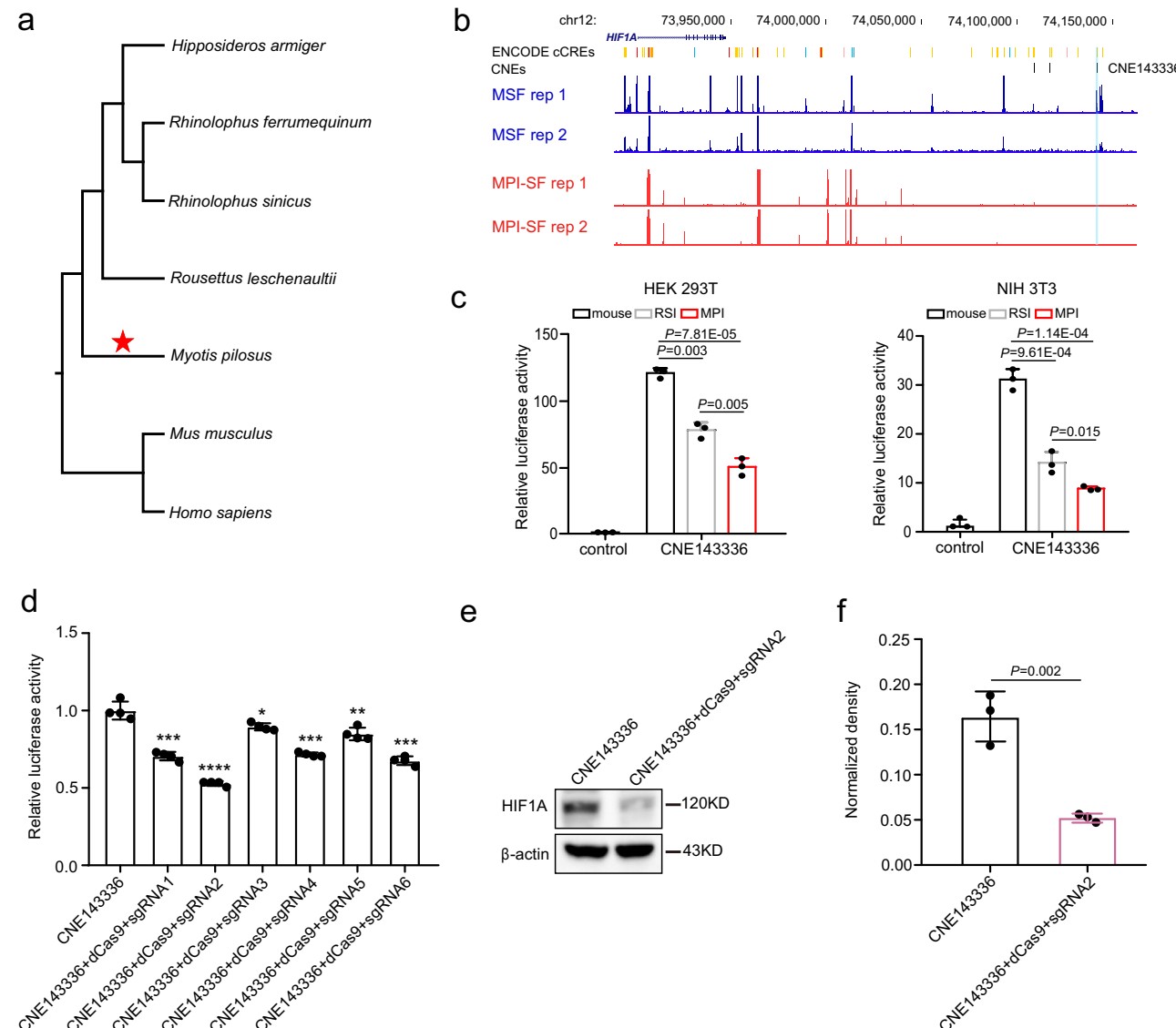

**Fig. 5 | A conserved non-coding element under accelerated evolution in MPI is associated with the downregulation of *HIF1A*. a** The phylogeny of seven species examined in this study is used to screen the evolutionarily conserved non-coding elements (CNEs) that are under accelerated evolution in MPI. **b** CNE143336 is located in the regions with ATAC-seq peaks in MSF and without ATAC-seq peaks in MPI-SF. Rep1 and 2 represent two biological replicates. **c** Luciferase reporter assay using HEK293T and NIH3T3 cell lines shows that the enhancer activity of CNE143336 is significantly lower in MPI compared to the orthologous sequences in mouse and the Chinese rufous horseshoe bat (RSI). The negative control is the PGL3-promoter empty vector-transfected in HEK293T and NIH3T3. **d** The regulatory activity of CNE143336 is blocked using CRISPR-dCas9 technology with guide RNAs derived from the sequence of CNE143336. **e, f** When the regulatory activity of CNE143336 is blocked, the expression level of *HIF1A* significantly decreases. The numbers of dots ($n = 3$) in (**c**), (**d**), and (**f**) represent the number of independent experiments. All data are presented as mean ± SD. The *P* values are from two-tailed Student's *t* tests. Source data are provided as a Source Data file.

their expression levels will be increased in MSF$^{HRAS\SV40LT}$, and unchanged or downregulated in MPI-SF$^{HRAS\SV40LT}$ compared to the corresponding MSF and MPI-SF without expressing HRAS(G12V) and SV40 LT. This is because MSF$^{HRAS\SV40LT}$ are susceptible to malignant transformation, while MPI-SF$^{HRAS\SV40LT}$ are resistant. We thus compared the protein expression levels of the three genes between MSF and MSF$^{HRAS\SV40LT}$, as well as between MPI-SF and MPI-SF$^{HRAS\SV40LT}$ (Fig. 6a). The results showed that RPS3 had similar protein levels between MSF and MSF$^{HRAS\SV40LT}$, as well as between MPI-SF and MPI-SF$^{HRAS\SV40LT}$ ($P > 0.05$; two-tailed Student's *t* tests; Fig. 6b). By contrast, the protein levels of HIF1A were significantly increased in both MSF$^{HRAS\SV40LT}$ and MPI-SF$^{HRAS\SV40LT}$ compared to their respective wild types ($P < 0.05$; two-tailed Student's *t* tests; Fig. 6b). Notably, the protein level of COPS5 was significantly increased in MSF$^{HRAS\SV40LT}$ ($P < 0.01$; two-tailed Student's *t*

test; Fig. 6b), but showed no significant difference in MPI-SF$^{HRAS\SV40LT}$ ($P > 0.05$; two-tailed Student's *t* test; Fig. 6b). In conjunction with our previous results, the downregulation of *HIF1A* and *COPS5* is more closely involved in the resistance of MPI cells to malignant transformation.

HIF1A and COPS5 have been shown to have a complex interplay. COPS5 not only directly interacts with HIF1A to increase its transcriptional activity, but also modulates HIF1A protein stability by competing with p53 or by inhibiting HIF1A prolyl-564 hydroxylation[26,27]. The regulatory relationship between HIF1A and COPS5 may be a key factor in the resistance of MPI cells to malignant transformation. Since HIF1A and COPS5 were co-upregulated in MSF$^{HRAS\SV40LT}$, we hypothesized that the transcription factor HIF1A could be responsible for the expression of *COPS5*, however, this regulation link may be disrupted in MPI. To

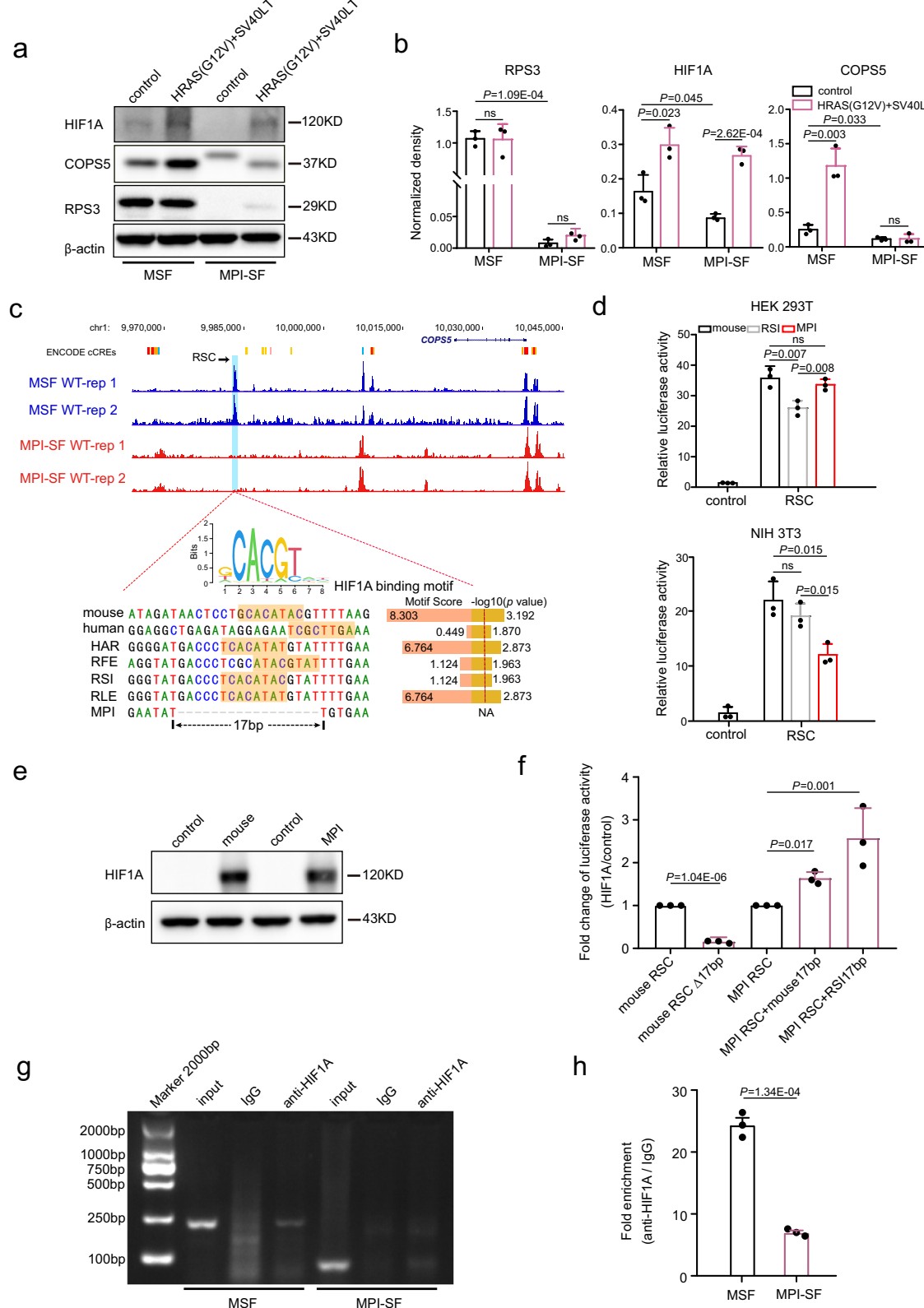

test this hypothesis, we conducted a screening of the genomic regions with ATAC-seq peaks in MSF but no peaks in MPI-SF around *COPS5*. By comparison, we identified a region located ~40 kb upstream of *COPS5* (Fig. 6c). This region, referred to as the potential regulatory sequence of *COPS5* (RSC), exhibited a 17 bp deletion specifically observed in MPI. Interestingly, this deletion was predicted to contain a binding site of HIF1A (Fig. 6c). To assess the enhancer activity of RSC, we performed

the dual-luciferase assay using both HEK 293T and NIH 3T3. Indeed, RSC from mouse, RSI, and MPI displayed distinct enhancer activity in the two cell lines compared to the negative control (Fig. 6d). In contrast to HEK 293 T, the enhancer activity of MPI RSC was significantly weaker than that of the RSCs from mouse and RSI in NIH 3T3 ($P < 0.05$; two-tailed Student's $t$ tests; Fig. 6d). NIH 3T3 is a type of fibroblast, and thus is probably more similar to the MPI fibroblasts examined in this

**Fig. 6 | The downregulation of *COPS5* in MPI may result from the loss of a potential enhancer. a** Immunoblotting is performed to analyze the protein levels of HIF1A, COPS5, and RPS3 in MSF, MSF^HRAS\SV40LT, MPI-SF, and MPI-SF^HRAS\SV40LT. **b** Comparison of the protein levels of HIF1A, COPS5, and RPS3 in fibroblasts with and without expressing oncogenic HRAS(G12V) and SV40 LT. **c** A putative enhancer containing a predicted HIF1A binding motif at the upstream of *COPS5* is specifically lost in MPI. The cyan box highlights the putative regulatory sequence of *COPS5* (RSC) with ATAC-seq peaks in MSF but without ATAC-seq peaks in MPI-SF. **d** Luciferase reporter assays are conducted using HEK293T and NIH3T3 cell lines to assess the enhancer activity of RSC from mouse, MPI, and RSI. The negative control is the PGL3-promoter empty vector-transfected in HEK293T and NIH3T3. **e** A cell model of HEK293T that continuously expresses *HIF1A* is generated. The experiment

was repeated independently three times with similar results. **f** The regulatory activity of mouse RSC significantly decreases when the 17 bp fragment containing the predicted binding motif is deleted. When the 17 bp fragments of mouse and RSI are respectively added to the RSC of MPI, the regulatory activity of the edited RSCs of MPI is significantly enhanced. **g** An antibody specific to HIF1A is used to enrich HIF1A along with its DNA targets in mouse and MPI fibroblasts, respectively. **h** ChIP-qPCR is performed using specific primers designed based on the RSC sequences. The results show that the concentration of mouse RSC is significantly higher than that of MPI RSC. The numbers of dots (n = 3) in (**b**), (**d**), (**f**), and (**h**) represent the number of independent experiments. All data are presented as mean ± SD. The *P* values are from two-tailed Student's *t* tests. ns: not significant. Source data are provided as a Source Data file.

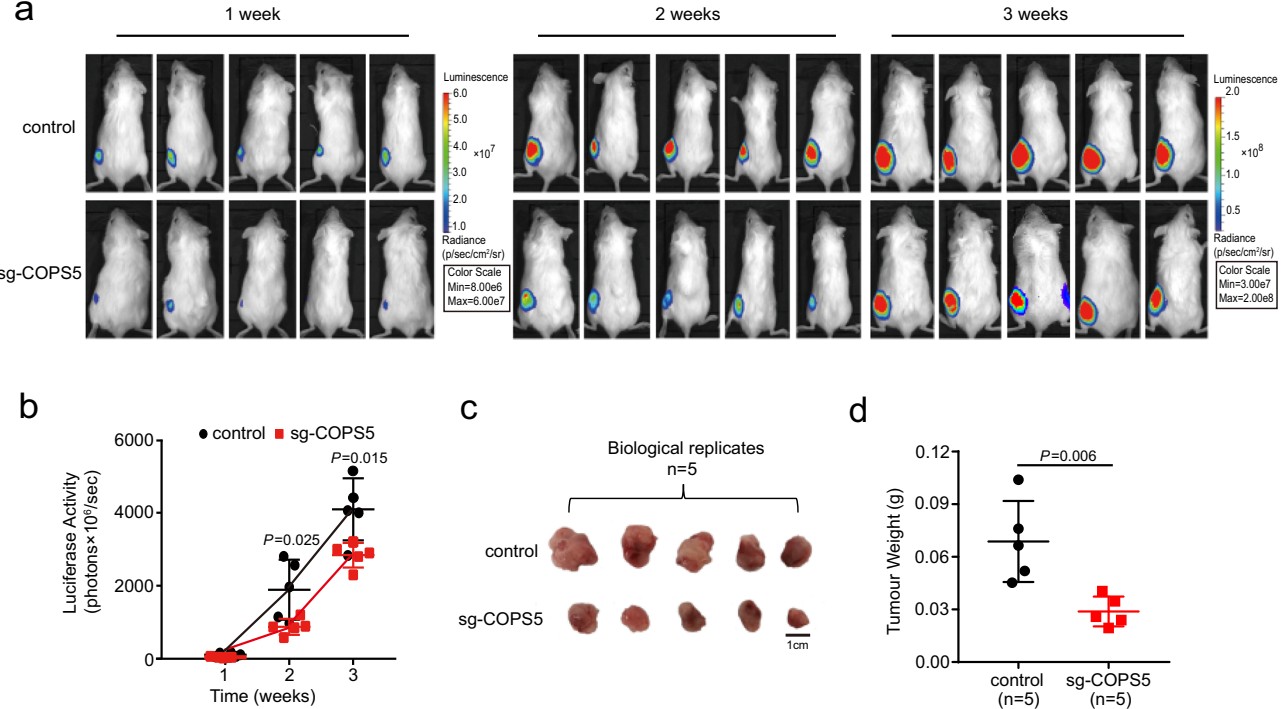

**Fig. 7 | The downregulation of *COPS5* reduces tumor size. a** Representative images of bioluminescence signals in immunodeficient mice bearing the xenografts derived from MSF^HRAS\SV40LT (control) and the COPS5-depleted MSF^HRAS\SV40LT (sg-COPS5) after 1, 2, and 3 weeks post subcutaneous implantation. **b** The luciferase activity is weaker for the xenografts derived from the COPS5-depleted MSF^HRAS\SV40LT than for those derived from MSF^HRAS\SV40LT 2 and 3 weeks post subcutaneous implantation. **c** Representative images of the xenografts respectively derived from

MSF^HRAS\SV40LT and the COPS5-depleted MSF^HRAS\SV40LT 3 weeks post subcutaneous implantation. Scale bar: 1 cm. **d** The xenografts derived from the COPS5-depleted MSF^HRAS\SV40LT are lighter than those derived from MSF^HRAS\SV40LT 3 weeks post subcutaneous implantation. The numbers of dots (*n* = 5) in (**b**) and (**d**) represent the number of biologically independent animals. All data are presented as mean ± SD. The *P* values are from two-tailed Student's *t* tests. Source data are provided as a Source Data file.

study. To confirm the binding ability of the predicted motif with HIF1A, we generated a stable cell line of HEK293T that continuously expresses *HIF1A* (Fig. 6e). Using this cell model, we conducted a dual-luciferase assay to assess the regulatory activity of RSC with or without this binding motif. When we deleted the 17 bp fragment containing this binding motif, the regulatory activity of mouse RSC significantly decreased (*P* = 1.04E-06; two-tailed Student's *t* test; Fig. 6f). Furthermore, when we respectively added the 17 bp fragments of the RSCs from mouse and RSI to the RSC of MPI, the regulatory activity of the edited RSCs of MPI significantly enhanced (*P* < 0.05; two-tailed Student's *t* tests; Fig. 6f). Additionally, we performed a ChIP-qPCR to demonstrate the binding ability of endogenous HIF1A to this binding motif. We used an antibody specific to HIF1A to enrich HIF1A along with its DNA targets in MSF and MPI-SF, respectively. After purification, we used specific primers designed based on the mouse and MPI RSC sequences around the 17 bp deletion to perform qPCR. Upon

comparison, we found that the concentration of this fragment of mouse RSC was significantly higher than that of MPI RSC (*P* < 0.001, two-tailed Student's *t* test; Fig. 6g, h). These results strongly support the presence of the binding site of HIF1A in the examined fragment of mice RSC but its absence in that of the MPI RSC sequence. Overall, our findings suggest that the loss of the HIF1A binding site upstream of *COPS5* may downregulate the expression of *COPS5* in MPI, consequently leading to the resistance of MPI cells to malignant transformation.

To further investigate whether the downregulation of *COPS5* is responsible for the resistance of MPI cells to malignant transformation, we established a cell-derived xenograft model as described above. We generated the COPS5-depleted MSF^HRAS\SV40LT using CRISPR-Cas9 gene-editing technology and subcutaneously injected them into immuno-deficient mice to form xenografts (Fig. 7a). After 2 and 3 weeks, the intensity of fluorescence was significantly higher in the xenografted

tumors derived from the COPS5-depleted MSF$^{HRAS\SV40LT}$ compared to the empty vector-transfected MSF$^{HRAS\SV40LT}$ ($P < 0.05$; two-tailed Student's $t$ test; Fig. 7b). Upon weighing the tumors 3 weeks post cell injection, we observed that the tumors derived from the COPS5-depleted MSF$^{HRAS\SV40LT}$ were significantly smaller and lighter than those derived from the empty vector-transfected MSF$^{HRAS\SV40LT}$ ($P = 0.006$; two-tailed Student's $t$ tests; Fig. 7c, d). Notably, the xenografts derived from the MPI-SF$^{HRAS\SV40LT}$ overexpressing *COPS5* were heavier than those derived from the MPI-SF$^{HRAS\SV40LT}$ 3 weeks post subcutaneous implantation ($P < 0.05$; two-tailed Student's $t$ test; Supplementary Fig. 6g). These results indicate a strong correlation between the decreased expression of *COPS5* and the inhibition of malignant transformation and tumor growth in vivo.

## Discussion

Laboratory mice and rats are traditionally utilized as model organisms to study carcinogenesis mechanisms due to their short lifespan, rapid reproduction, and cancer susceptibility. However, these cancer-prone model organisms have limitations in understanding the mechanisms of cancer resistance. In contrast, certain non-model organisms, such as naked mole rats[13,28], blind mole rats[12,29], and elephants[30], have been shown to possess cancer resistance. These species have evolved unique anticancer mechanisms[6], suggesting that the identification of species resistant to cancer can provide valuable insights into natural cancer resistance processes. In this study, we offered experimental evidence, both in vitro and in vivo, for the malignant transformation resistance of MPI cells, strongly implying that the big-footed bat (*Myotis pilosus*) is a mammalian species resistant to cancers. It is important to note that while the oncogenic HRAS(G12V) is the most common cancer-associated substitution and is also commonly used to induce malignant transformation of cells[11], particularly in non-model organisms like naked mole-rats and blind mole-rats[12,13], neoplastic transformation of cells may require species- or cell type-specific mutations of HRAS or other RAS members[10,11]. Therefore, our results cannot completely rule out the possibility that the other six bat species examined in this study may have evolved different anticancer mechanisms compared to MPI. Overall, our findings not only directly corroborate the hypothesis that bats have evolved cancer resistance[5,6,8,20], but also suggest that the cancer resistance may have evolved independently in various bat lineages[31,32].

Why has MPI evolved cancer resistance? The most likely scenario is that it is a long-lived bat species, as there is a close mechanistic relevance between cancer resistance and longevity[5,6]. MPI belongs to the bat genus *Myotis*, which includes the oldest recaptured bat, *Myotis brandtii*, that lived ~8 times (>41 years old) longer than expected for its size[33]. There are also at least 13 species in this genus that live more than 20 years and four species that live more than 30 years, despite their small body sizes[2,34]. Unfortunately, there is no record of the lifespan of MPI. However, it is worth noting that cells from long-lived species tend to undergo senescence more than cells from short-lived species[35]. Therefore, we induced senescence in the primary cell lines of the eight species mentioned above using etoposide. When examining the commonly used senescence indicator, we found that the β-galactosidase activity was significantly higher in MPI fibroblasts compared to the other six bat species, as well as mouse ($P < 0.01$; two-tailed Student's $t$ tests; Supplementary Fig. 8a, b). Additionally, the protein levels of p21 and p53, which are marker genes for cellular senescence[36,37], were significantly higher in MPI fibroblasts than in mouse fibroblasts ($P < 0.01$; two-tailed Student's $t$ tests; Supplementary Fig. 8c, d). These findings suggest that MPI may have a longer lifespan than the other six bat species examined, as well as mice. Interestingly, previous studies have shown that the depletion of *COPS5* and *HIF1A*, as observed in MPI, leads to premature senescence[38–40]. Cellular senescence is widely believed to be an important mechanism against cancer[37,41], indicating that MPI may have evolved to have a propensity for cellular senescence, which is associated with both longevity and cancer resistance.

Bats are known to have genomes that contain many active transposon elements (TE)[42]. Therefore, we investigated whether the loss of the potential enhancer containing the binding site of HIF1A is related to TE activity in MPI. We detected TEs in the MPI genome using a previous method[42] and found a total of 3,469,596 TEs (Supplementary Data 8). Among the CNEs that showed accelerated evolution in MPI, we identified 1182 that overlapped with these TEs. Its ratio was significantly lower than that of the CNEs that did not show accelerated evolution in MPI overlapped with TEs ($P < 0.001$; two-tailed $\chi^2$ test), suggesting that the accelerated evolution of CNEs in MPI was unrelated to transposon activity. Nevertheless, we did identify 13 and 31 TEs within 5 kb and 10 kb fragments around the loss of the potential enhancer, respectively. Thus, we cannot rule out the possibility that the loss of the enhancer is related to TE activity.

It is noteworthy that long-lived mammalian species, which are naturally resistant to cancers, have evolved diverse tumor suppressor mechanisms. For instance, the naked mole rats exhibit early contact inhibition in their fibroblasts, which is caused by the activation of p16 and high molecular mass hyaluronan[13,28]. The blind mole rat's fibroblasts undergo a combination of necrotic and apoptotic processes of cell death mediated by a large release of IFNβ[29]. In the case of MPI, the decreased expression of *HIF1A* and *COPS5* may represent an alternative anticancer mechanism. Indeed, *HIF1A* is known to be overexpressed in different cancer progressions and activates the transcription of genes involved in crucial aspects of cancer biology, such as angiogenesis, cell survival and proliferation, glucose metabolism, and invasion[43,44]. Similarly, *COPS5* has been observed to be amplified in several carcinomas and plays an important role in tumorigenesis by interacting with various oncogenic genes, including p53 and p27[45,46]. The downregulation of *HIF1A* and *COPS5* has been shown to inhibit the proliferation and malignant behaviors of cancer cells, suggesting that inhibitors targeting the two proteins may have potential as anticancer therapeutics[43,44,47,48]. Our findings suggest that HIF1A may promote the expression of *COPS5*. However, it is also possible that COPS5 may enhance the stability and abundance of HIF1A, as it is known to stabilize HIF1A[27]. Indeed, when inhibiting the expression of *COPS5*, we observed a significant decrease in *HIF1A* expression ($P < 0.01$, two-tailed Student's $t$ test; Supplementary Fig. 9). In addition, although RPS3 has very low expression levels in MPI-SF and MPI-SF$^{HRAS\SV40LT}$, we cannot completely disregard its role in the resistance of MPI cells to malignant transformation. This is because the overexpression of *RPS3* significantly increased cell proliferation (Fig. 4d), and when *HIF1A*, *COPS5*, and *RPS3* were overexpressed together in MPI cells, it resulted in the formation of significantly larger soft agar colonies and tumors (Supplementary Fig. 6).

Given that most cancers in humans arise from epithelial cells, rather than fibroblasts, to further explore the relationship between the downregulation of *COPS5* observed in fibroblasts and tumor cells, we inhibited the expression of *COPS5* in human proximal tubular epithelial cells (HK-2), a breast cancer cell line derived from mammary epithelium (MCF-7), and a pancreatic cancer cell line derived from the pancreatic duct epithelium (PANC-1). The results showed that these epithelial and tumor cell lines exhibited a significant decrease in proliferation ($P < 0.01$, two-tailed Student's $t$ tests; Supplementary Fig. 10), providing further evidence for the important roles of *COPS5* downregulation in the resistance of MPI cells to malignant transformation. Furthermore, we observed that the protein size of COPS5 was larger in MPI than in other species, but when MPI fibroblasts expressed HRAS(G12V) and SV40LT, its size became comparable to that of other species (Fig. 6a). To confirm that this difference was not due to nonspecificity of the COPS5 antibody, we inhibited the expression of *COPS5* in MPI cells and found that the band intensity was significantly weakened ($P < 0.01$, two-tailed Student's $t$ tests; Supplementary

Fig. 11). This result suggests that the observed difference in protein size did not result from the non-specificity of the COPS5 antibody and that alternative splicing or certain chemical modifications may be involved in the regulation of MPI *COPS5*, despite its lack of involvement in the resistance of MPI cells to malignant transformation. In addition, it is important to note that the size of the tumors derived from *COPS5*-depleted MSF[HRAS\SV40LT] did not decrease to the same extent as those derived from the MPI-SF[HRAS\SV40LT] (Figs. 2c, 7c). This suggests that the downregulation of *COPS5* may only represent a tip of the iceberg of anticancer mechanisms in MPI and that there may be other factors contributing to the resistance of MPI cells to malignant transformation.

While laboratory mice have been extensively used as a research model for human cancers and have provided valuable insights, it is important to acknowledge that there are substantial differences in carcinogenesis between humans and mice. Unlike cancer-prone mice, human cells are more resistant to malignant transformation[10,49], a phenomenon that is also observed in other long-lived mammalian species. Consequently, investigating the anticancer mechanisms in long-lived mammalian species has the potential to yield ground-breaking advancements in cancer treatment and prevention for humans.

## Methods
### Ethical regulations statement
All animal care and experimental protocols were approved by the Institutional Animal Care and Use Committee of the Kunming Institute of Zoology, Chinese Academy of Sciences (IACUC-PA-2021-06-013) and in accordance with the Animal Research: Reporting of In Vivo Experiments guidelines.

### Samples and cell culture
Due to the sample availability, we obtained primary skin fibroblasts from seven common bat species in the southwest region of China, including the big-footed bat (*Myotis pilosus*, MPI for short), the Szechwan myotis (*Myotis altarium*, MAL), the least horseshoe bat (*Rhinolophus pusillus*, RPU), the greater horseshoe bat (*Rhinolophus ferrumequinum*, RFE), the Chinese rufous horseshoe bat (*Rhinolophus sinicus*, RSI), the great leaf-nosed bat (*Hipposideros armiger*, HAR), and the Leschenault's Rousette (*Rousettus leschenaultii*, RLE), as well as the laboratory mouse (*Mus musculus*). These bat fibroblasts were available from the CAS Kunming Cell Bank in the Kunming Institute of Zoology, Chinese Academy of Sciences. These cell lines were cultured in DMEM media (Gibco) supplemented with 10% fetal bovine serum (Gibco) at 5% $CO_2$, 3% $O_2$, and 37 °C. All cell lines were used at an early passage (<10 population doublings).

### Lentivirus preparation and the generation of stable cell lines
To enhance the stability and repeatability of our experiments, we created stable cell lines expressing oncogenic HARS(G12V) and SV40 Large T (SV40 LT) for each of the eight mammalian species. We constructed the lentiviral plasmid PCDH-CMV-HRAS(G12V)-SV40 LT by combining PSG5 Large T (Addgene#9053), PCDH3-HRAS(G12V) (Addgene#39504), and the backbone plasmid PCDH-CMV-MCS. The plasmid PCDH-CMV-HRAS(G12V)-SV40 LT and the packaging plasmids pSPAX2 and pMD2.G were co-transfected into HEK 293T cells using Lipofectamine 3000 (Thermo Fisher) at a ratio of 10:5:2. After 48 and 72 h of transfection, the lentivirus particles were harvested twice and filtered using a 0.45 µm filter (Millipore). Primary cells were then infected with the lentiviruses and selected with 2 µg/ml puromycin for 1 week to generate stable cell lines expressing HARS(G12V) and SV40 LT.

### Protein lysates and immunoblotting
When the cells stably expressing HARS(G12V) and SV40 LT from eight species reached ~90% confluence, they were scraped off and washed twice with PBS. The cells were then lysed in RIPA buffer (Beyotime) containing a protease inhibitor cocktail (Bimake) and centrifuged to remove any debris. Standard SDS-PAGE gel electrophoresis was performed, followed by blocking with 10% skimmed milk. The membranes were then incubated with primary antibodies overnight at 4 °C. The primary antibodies used in this experiment were anti-HRAS(G12V) (1:500; CST), anti-SV40 LT (1:500; Sant-Cruz), and anti-β-actin (1:5000; Sigma-Aldrich). The images were captured using a chemiluminescence image analysis system (Tanon 5200), and the density of the protein bands was quantified using Image-pro plus 6.0 software.

### Anchorage-independent soft agar growth assay
A single standard six-well plate was prepared by diluting the stock agar solution with a growth medium. The mixture was then cooled at 4 °C for ~5 min to allow the 1 ml basal agarose layer (0.8%) to solidify. For each well, a total of $2 \times 10^4$ cells were mixed with 1 ml complete medium containing 0.4% soft agar. The mixture was then added to the top of the solidified base agar. After cooling at 4 °C for 5 min, 2 ml complete medium was added to each well. The plates containing the soft agar and cells were incubated at 37 °C and 5% $CO_2$ for 4 weeks. After the incubation period, the plates were photographed under an inverted microscope at 10× and 20× magnifications.

### Cell-derived xenograft model
Five-week-old B-NDG (NOD. CB17-Prkdc[scid]Il2rg[tm1]/Bcgen) mice were purchased from Jiangsu Biocytogen Co., Ltd (Nantong, China) and were housed in a pathogen-free environment for 1 week before the start of the experiment. Fibroblasts stably expressing HRAS(G12V) and SV40 LT were infected with a lentivirus expressing Luciferase and were selected with 8 µg/ml blasticidin for 7 days. Approximately $1 \times 10^6$ cells in 100 µL normal saline containing 30% Matrigel (Corning) were then subcutaneously injected into each mouse. Weekly bioluminescent imaging of xenograft growth in vivo was conducted by intraperitoneally injecting mice with 150 mg/kg D-Luciferin (Beyotime) and photographing them using the IVIS system. The images were then analyzed using the Living Image software (Caliper Life Science, IVIS Lumina Xr). Upon reaching a tumor volume of ~2 $cm^3$ and did not exceed the maximal tumor size allowed by the ethical approval, the mice were sacrificed, and the tumors were weighed.

### Transcriptome sequencing and analyses
Total RNA was extracted from fibroblasts of the eight mammalian species mentioned above (mouse, MPI, MAL, RPU, RFE, RSI, HAR, and RLE) using QIAzol Lysis Reagent (Qiagen). To ensure reliable and repeatable results, three biological replicates were conducted. The quality and integrity of the purified total RNA were assessed using the Agilent 2100 Bioanalyzer (Thermo Fisher) and the RNeasy Plus Universal Mini Kit (Qiagen). Approximately 1 µg of total RNA from each sample was used to construct cDNA libraries following the manufacturer's instructions. The libraries were then sequenced on the MGISEQ 2000 platform (BGI-Shenzhen, China) in a paired-end form with 150 bp. Raw reads obtained were trimmed using Trim Galore (www.bioinformatics.babraham.ac.uk/projects/trim_galore) with the following parameters: --phred33 -q 25 --length 35 --stringency 4 -e 0.1. The resulting clean data was then assembled with Trinity (v2.8.5)[50]. The expression level of each transcript was calculated using the Trinity Script, align_and_estimate_abundance.pl. The expression levels were then quantified as expected fragments per kilobase of transcript per million fragments (FPKM). Integration of one-to-one orthologous genes across the eight species resulted in a total of 6314 genes with FPKM > 0.5 in more than half of the 24 samples. These genes were selected for weighted gene co-expression network analysis. The raw sequence data have been deposited in the Science Data Bank (https://doi.org/10.57760/sciencedb.08307) and in the Gene Expression Omnibus at the National Center for Biotechnology Information (PRJNA976519).

## Weighted gene co-expression network analysis

The *R* package of signed Weighted Gene Co-expression Network Analysis (WGCNA)[15] was utilized to construct gene co-expression networks. The FPKM values of the 6314 one-to-one orthologous genes across 24 samples from the eight species were logarithmically transformed ($\log_2(\text{FPKM} + 1)$). This transformed gene expression matrix was used to construct the co-expression network using WGCNA. To determine the appropriate soft threshold for network construction, various soft threshold values were tested (Supplementary Fig. 12). Based on the results, a soft threshold of 12 was selected. The network was constructed with the following parameters: power = 12, networkType = "signed", TOMType = "signed", minModuleSize = 30, reassignThreshold = 0, mergeCutHeight = 0.25. This resulted in the identification of 34 co-expression modules, which were numerically labeled from 1 to 34.

## Protein–Protein Interaction (PPI) network analysis

A PPI network was constructed using 210 genes from the upregulated module M1 and 146 genes from the downregulated module M2. The interactions between these genes were obtained from the STRING database[16]. The NetworkAnalyzer tool of Cytoscape (version 3.7.2)[51] was used to calculate the betweenness centrality and degree of each of the 178 nodes in the network.

## Prognosis of cancer patients

The prognostic data for each gene from the gene co-expression modules M1 and M2 across 21 tumor types were collected from the KMplot database[18]. The analysis involved counting the number of tumor types in which the gene expression level significantly affected patient survival ($P < 0.05$) for all genes in M1 and M2. Next, the dataset was analyzed to determine the number of tumor types in which higher or lower expression levels of the genes in the upregulated module (M1) and in the downregulated module (M2) significantly prolonged patient survival, respectively. Finally, a frequency distribution was generated to represent the proportions of these different tumor types relative to the total number of tumor types with significant survival rates for the genes in both M1 and M2.

## CRISPR/Cas9-mediated genome editing

The single-stranded guided RNA sequences (sgRNA) for each of the five genes (*HIF1A*, *COPS5*, *RPS3*, *EIF5B*, and *EP300*) were designed to induce nucleic acid cleavages at coding regions, as shown in Supplementary Data 9. Each sgRNA was cloned into the lentiCRISPRv2-hygro plasmid (Addgene #98291) using Esp3I digestion (Thermo Fisher). To confirm the correctness of the cloned sgRNAs, single colonies of cells were amplified, and their sequences were verified through Sanger sequencing. Stable cell lines were generated and the efficiency of gene knock-out was evaluated by immunoblotting. Immunoblotting was conducted using primary antibodies specific to each target protein: anti-HIF1A (1:500; ZENBIO), anti-COPS5 (1:1000; CST), anti-RPS3 (1:500; ZENBIO), anti-EIF5B (1:500; Affinity), and anti-EP300 (1:500; ZENBIO).

## Cell proliferation assay

The protein-coding sequences of *HIF1A*, *COPS5*, and *RPS3* from MPI were synthesized and cloned into the pLenti CMV GFP-Hygro (Addgene #17446), respectively. Stable cell lines were constructed and the efficiency of gene overexpression was assessed using immunoblotting. To evaluate the effect of gene expression on cell proliferation, a cell proliferation assay was performed using an MTT reagent following the manufacturer's protocol (Promega). Cells were counted and seeded into 96-well plates at a density of 3000 cells per well. Every 24 h, a mixture of 90 μL DMEM media and 10 μL MTT reagent was added to each well and incubated for 1 h at ambient temperature. The absorbance value was measured at 450 nm using a Hybrid reader (Synergy H1).

## Genome sequencing and assembly

To identify the evolutionarily conserved non-coding elements that underwent accelerated evolution in MPI, we obtained different fresh tissues of a male MPI (provided upon request) to sequence its genome with high quality. Genomic DNA was extracted from the muscle tissue, and its concentration, integrity, and purity were assessed. Long fragments of the genomic DNA were selected using the BluePippin system (Sage Science) and sequenced on the Nanopore GridION X5/PromethION platform (Oxford Nanopore Technologies). The total amount of clean data was 206.4 Gb, with 10,201,807 reads of an average length of 20.2 Kb, an N50 of 28.1 Kb, and the longest read length of 651.2 Kb. The genome was assembled using the NextDenovo software[52]. The reads_cutoff and seed_cutoff parameters were set to 1k and 26k, respectively. Briefly, the NextCorrect module was used to self-correct the original subreads to obtain consistent sequences (CNSs). The CNSs were then de novo assembled using the NextGraph module with default parameters to construct a preliminary genome draft. Finally, the Nextpolish module was utilized to refine the genome draft by combining Nanopore long-read sequencing data, Pb HiFi data, and second-generation sequencing data. The BUSCO[53] was employed to evaluate the completeness of the assembled genome, with 8888 (96.3%) of the 9226 single-copy orthologs genes for the assembled MPI genome.

## Annotation of repetitive elements in the MPI genome

Following our previous study[54], we employed RepeatModeler (v1.0.4)[55] to generate a de novo repeat sequence library from the MPI genome. This library was then combined with the mammalian repeat sequence library from the RepBase database[56] to create a comprehensive repeat sequence library. Using this comprehensive repeat library, RepeatMasker[57] was used to retrieve the repetitive elements in the MPI genome. This analysis revealed that ~41.04% of the genome was identified as repetitive elements and subsequently masked.

## Annotation of the MPI genome

The Broad Institute eukaryotic annotation pipeline[58] was utilized to predict protein-coding genes in the MPI genome, which included RNA-sequencing (RNA-seq)-based gene prediction, homology-based gene prediction, and ab initio gene prediction. For the RNA-seq-based gene prediction, transcriptomic data were initially obtained from seven distinct MPI tissues (heart, liver, spleen, lung, kidney, brain, and cochlea). These transcriptome reads were de novo assembled using Trinity[50]. The assembled genomes were then used to facilitate the clustering of transcriptome reads, and the final transcripts were constructed from the transcriptome reads. PASA[59] was applied to integrate the assembled transcriptomes to generate a consensus gene set based on overlapping transcript alignment. For homology-based gene prediction, nine high-quality genomes were selected, including human (*Homo sapiens*), mouse (*Mus musculus*), dog (*Canis familiaris*), the flying fox (*Pteropus vampyrus*), the greater horseshoe bat (*Rhinolophus ferrumequinum*), the little brown bat (*Myotis lucifugus*), the greater mouse-eared bat (*Myotis myotis*), the Natal long-fingered bat (*Miniopterus natalensis*) and the Kuhl's pipistrelle (*Pipistrellus kuhlii*) (Supplementary Data 10). TblastN was used to align all protein sequences from each species to the assembled genome, and GeMoMa[60] was then used to predict gene structures. The ab initio gene prediction of gene models based on the masked assembled genome was conducted using GlimmerHMM[61], geneid[62], and genscan. The above three gene prediction datasets were integrated using the EVM[58] to generate a comprehensive and nonredundant gene set. Finally, PASA was applied again to further update the EVM consensus predictions by adding untranslated region annotations and gene models for alternatively spliced isoforms. To ensure a high-quality set of protein-coding genes, we only retrieved those that met the following criteria: (a) the length of the encoded amino acid sequences was ≥50; (b) the encoded protein sequences

could be mapped to the National Center for Biotechnology Information (NCBI) nonredundant protein database with a cutoff of $E < 10^{-5}$; (c) the ratio of the alignment length to the query length was ≥0.4; and (d) the ratio of the alignment length to the subject length was ≥0.4. By applying these criteria, a total of 20,742 protein-coding genes were obtained. This number is highly comparable to results from other mammals with high-quality genomic data.

## Whole-genome alignment

Whole genome alignment was conducted for seven species examined in this study, including the laboratory mouse, the human, the big-footed bat, the greater horseshoe bat, the Chinese rufous horseshoe bat, the great leaf-nosed bat, and the Leschenault's Rousette. The genome resources and versions for these species are provided in Supplementary Data 11. The mouse genome (version mm10) was used as the reference genome for performing pairwise and multiple genome alignments. Lastz was employed with the following parameters ($K = 3000$ $L = 3000$ $Y = 9400$ $E = 30$ $H = 2000$) to construct pairwise genome alignments between the mouse and each of the six query species. Then, axtChain[63] was used to compute co-linear alignment chains based on the pairwise alignments. These alignment chains represent regions of conserved sequence between the genomes. The pairwise alignment chains were further processed using a modified version of chainNet[63], which calculates real scores of partial nets. This conversion step resulted in the generation of alignment nets, which provide a more comprehensive representation of the conserved regions across the genomes. To analyze the evolutionary pressures on the branch of MPI (*Myotis pilosus*), the protein-coding sequences of *HIF1A*, *EP3OO*, *EIF5B*, *COPS5*, and *RPS3* were collected across these seven species. These sequences were then aligned using PRANK[64] for each gene. To estimate the evolutionary pressures specifically on the branch of MPI, the branch-site model of Codeml, which is built in PAML[19], was utilized. This model allows for the detection of positive selection acting on specific lineages or branches in the phylogenetic tree.

## Identification of evolutionarily conserved non-coding elements

Following a previous study[65], the PHAST msa_view was used to extract four-fold degenerated codon positions based on the mouse genome annotation. Next, PHAST PhyloFit was employed to estimate the length of all branches in the phylogenetic tree as substitutions per neutral site, which was used to detect evolutionarily conserved CNEs with PHAST PhastCons under the parameters: expected-length = 45, and target-coverage = 0.3. Subsequently, PHAST phyloP−ACC was employed to detect CNEs under accelerated evolution specifically on the branch of MPI. Only CNEs with significantly accelerated signals with a Benjamini–Hochberg false discovery rate ≤0.05 were considered.

## ATAC-seq data analysis

The lysis buffer (10 mM Tris-HCl, pH = 7.4, 0.1% Tween-20, 0.1% Nonidet P40 Substitute, 0.01% Digitonin, 0.1% BSA, 1 mM DTT, 10 mM NaCl, 1 U/μl Rnase inhibitor, 3 mM MgCl$_2$) was used to grind cells for each sample and ~$5 \times 10^4$ nuclei were distributed for each library. These nuclei were suspended using 50 μl transposition reaction MIX, which was then placed in a PCR machine at 37 °C for 30 min to amply the library. AMPure XP beads (Beckman) were used to purify the amplified library and the Qubit was used to quantify the concentration of the library. Finally, ATAC-seq libraries were sequenced on the Illumina Novaseq 6000 platform with 150 bp paired-end reads.

The ENCODE pipeline (github.com/ENCODE-DCC/atac-seq-pipeline) (version 1.8.0) was employed to process ATAC-seq data. Briefly, the cutadapt with default parameters was used to trim primer sequences, and Bowtie2[66] (version 2.3.4.3) was used to align the trimmed reads to the mouse genome (version mm10) and the MPI genome,

respectively. After removing mitochondrial reads, the sorted and indexed binary alignment mapped files were generated using Samtools (version 1.9)[67]. PCR duplicates were removed from the bam files using Picard (broadinstitute.github.io/picard) (version 2.20.7), and BEDTools[68] (version 2.29.0) was used to remove mm10 blacklisted regions (storage.googleapis.com/encode-pipeline-genome-data/mm10/mm10.blacklist.bed.gz). The number of reads and fragment length distribution in bam files were determined using Samtools. Peak-calling for ATAC-seq was performed using MACS2 (version 2.2.4)[24]. ATAC-seq peaks derived from IDR analysis of biological replicates were lifted between species using the halliftOver (version 2.2)[69] based on the genome alignment of mouse and MPI. The ATAC-seq data was visualized using Integrative Genomics Viewer (version 2.8.7)[70] after RPKM normalization and bigwig conversion was conducted using wigToBigWig (ucsc-wigtobigwig ==357)[71]. The UCSC-provided ENCODE cCREs track[72] was also visualized.

## Analysis of transcription factor binding motifs

To identify the binding motifs of transcription factors within the CNEs, the motifs were derived from PECA2[73]. The Homer[74] tool was then used to predict the motifs based on the provided data. To determine the presence of motifs in each CNE sequence of each species the motifs were scored using FIMO[75] and our in-house code[76]. If the score was significantly greater than 0, it was assumed that the CNE contained the binding motif.

## Luciferase reporter assay

The candidate CNEs were synthesized and cloned into the pGL3-promoter vector. Approximately $2 \times 10^4$ HEK 293T cells were seeded into each well of a 96-well plate. Co-transfection of 150 ng of the plasmid (pGL3-promoter, pGL3-CNE143336) driving the firefly luciferase and 30 ng of the control plasmid (pRL-TK) expressing renilla luciferase was conducted using Lipofectamine 3000 (Thermo Fisher). After 48 h, luciferase activity was measured following the manufacturer's protocol (Promega). Three replicates were used for the experiments. Relative enhancer activity was determined by normalizing the mean of the replicates of the empty control. This procedure was repeated using the NIH 3T3 cell line.

## Regulatory activity inhibition assay

We used the dCas9-KRAB and single guide RNA (sgRNA) system to decline the regulatory activity of CNE143336. Briefly, HEK 293 T cells were co-transfected with the following plasmids using Lipofectamine 3000 (Thermo Fisher): dCas9-KRAB plasmid (200 ng), pGL3-CNE143336 driving the firefly luciferase (200 ng), control plasmid pRL-TK (30 ng), and one of sgRNAs (sgRNA1-sgRNA6) targeting the sequence of mouse CNE143336 (600 ng). The specific sgRNA sequences can be found in Supplementary Data 12. After 48 h of transfection, luciferase activity was measured using the luciferase assay system. Stable cell lines expressing dCas9-KRAB and sgRNA2 were constructed in NIH 3T3 cells. Immunoblotting was performed to detect the protein level of endogenous HIF1A in the stable NIH 3T3 cell lines expressing dCas9-KRAB and sgRNA2.

## Chromatin immunoprecipitation (ChIP)-qPCR assay

ChIP assays were conducted following the manufacturer's protocol (Beyotime). In brief, ~$4 \times 10^7$ fibroblasts were cross-linked with 1% (w/v) formaldehyde for 10 min at 37 °C. The cross-linking reaction was then stopped by adding 1.1 ml glycine. The cross-linked cells were sonicated using a sonicator with 25% amplitude and 30-s bursts with a 40-s reprieve, repeated six times. This sonication step aimed to fragment the DNA into fragments ranging from 200 to 800 bp. The sonicated DNA was immunoprecipitated using anti-HIF1A (CST) or a control rabbit IgG (Beyotime). The immunoprecipitated DNA was purified with the QIAquick PCR Purification Kit (QIAGEN). Using the purified ChIP

**Article**

DNA as templates, the relative occupancies of the mouse and MPI RSCs were quantified using GoTaq® qPCR master mix (Promega) and their specific primers (Supplementary Data 13).

### SA-β-gal staining

Fibroblasts from the seven species above were seeded on a six-well plate at 50% confluence. Then, the cells were treated with 10 µmol/L etoposide (Sigma-Aldrich) for 12 h. After 7 days, SA-β-gal staining was conducted according to the manufacturer's instructions (Beyotime). Three distinct areas, each containing a minimum of 100 SA-β-gal-stained cells, were photographed for each sample.

### Statistics and reproducibility

Statistical analyses were conducted using GraphPad Prism (GraphPad, USA). The two-tailed Student's $t$ tests were employed to determine statistical differences between two groups. A significance level of $P < 0.05$ was considered statistically significant, $P < 0.01$ was deemed highly significant, and $P < 0.001$ was considered extremely significant. The groups being compared exhibited similar variances. The sample size was not predetermined using statistical methods, and no data were excluded from the analyses. For all animal experiments, mice of matching age, gender, and background were randomly assigned to the respective groups. The investigators were not blinded to group allocation during the experiments or outcome assessment. The data are presented as means ± standard deviation (SD).

### Reporting summary

Further information on research design is available in the Nature Portfolio Reporting Summary linked to this article.

## Data availability

All data generated or analyzed during this study are included in the article and its Supplementary Information. The sequencing data generated in this study were deposited in the NCBI database, including RNA-seq (PRJNA976519) and ATAC-seq (PRJNA975438). This Whole Genome project has been deposited at GenBank under the accession JASKON000000000 (PRJNA975353). The version described in this paper is version JASKON010000000. These sequencing data have also been deposited in the Science Data Bank [https://doi.org/10.57760/sciencedb.08307]. Source data are provided with this paper.

## Code availability

Custom code is available on Zenodo at https://doi.org/10.5281/zenodo.10262324[76].

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

## Acknowledgements

We thank Guo-Dong Wang for his valuable comments. This work was funded by grants from the National Key R&D Program of China (2022YFC2602500), the National Natural Science Foundation of China (32192422, U23A20452, 32330014), the Yunnan Fundamental Research Projects (202201AS070058, 202102AA310055), the CAS "Light of West China Program" (xbzg-zdsys-202113), and the Key Research Program of the Chinese Academy of Sciences (KJZD-SW-L11).

## Author contributions

Z.L. conceived and supervised the project. R.H., L.Y., L.-Y.S., X.-Q.Y., and H.-Y. Z. performed experiments. Z.L. provided tissue samples of MPI for sequencing. Q.-Y.H. annotated the MPI genome and identified conserved non-coding elements. J.-J.H. conducted gene-coexpression analyses. Y.-S.M. performed transcriptome and ATAC-seq analyses. Z.L. and R.H. wrote the paper with input from all authors.

## Competing interests

The authors declare no competing interests.
