## [Peer Review File · Nature Communications]

REVIEWER COMMENTS

Reviewer #1 (Remarks to the Author): Expert in comparative cancer genomics, cancer and evolution in mammals, and epigenomics

The authors have investigated cancer resistance in seven bat species by activating oncogenic Ras and SV40 LT in their fibroblasts, and found the bat species *Myotis pilosus* (MPI) is the most resistant. Via comparative transcriptome, they have discovered that three genes HIF1A, COPS5, and RPS3 are downregulated in MPI. They further report that a potential enhancer containing the HIF1A binding site upstream of COPS5 is lost in MPI, and the inhibition of COPS5 expression reduces the xenograft tumor formation in mice. Their work sheds light on the molecular mechanisms of cancer resistance mechanisms in bats, some of which are long-lived species. The manuscript is well-written.

The following issues need to be addressed:

1. Does MPI has the longest life span among the 7 species compared?
2. What is the rationale using fibroblasts to do the experiment? Most cancers in humans arise from epithelial cells (adenomas or carcinomas), and fibroblasts are among the cell types that consist of tumor microenvironment. How the molecular changes in fibroblasts influence tumor cells should be discussed. Will using fibroblast compromise the interpretation of the survival results in Figure 4B?
3. It would be informative to inhibit the expression COPS5 in human epithelial cells to see what happened.
4. Bats are known for their genomes harboring many active transposons. Is the loss of the potential enhancer related to transposon insertion/deletion? Is the accelerated evolution of 5,938 CNEs in MPI related to transposon activity?

Minor issues

1. Figure 4B is hard to understand: more explanation should be provided in the Figure legends.
2. Genes should be in italics.

Reviewer #2 (Remarks to the Author): Expert in cancer molecular and functional biology, and cancer in bats

This is an interesting study aimed at identifying bat species that are resistant to cancer development and exploring the underlying mechanisms of their resistance. The authors demonstrated that out of eight bat

species, *Myotis pilosus* (MPI) cells exhibited particularly strong resistance to cancer. In comparison to other species and mouse cells, MPI cells demonstrated a strong resistance to transformation caused by oncogenic Ras and SV40 LT. Through a combination of bioinformatics and transcriptomics analyses, they revealed that HIF1A, COPS5, and RPS3 are downregulated in MPI cells and that depletion of these genes in mouse cells reduces proliferation. Additionally, the study showed that the HIF1A binding motif located upstream of the COPS5 gene is lost in MPI cells. They interpreted that MPI cells fail the HIF1A-mediated COPS5 induction after the introduction of Ras and SV40 LT, which may prevent cancer progression. In line with this, they demonstrated that knockout COPS5 in mouse cell lines reduced their tumor formation in mice. Some mammals have unique ways of preventing tumors. This is a growing area of interest, as we may be able to learn from these strategies to develop new cancer preventions/treatments for humans. Therefore, their findings may be very interesting in this field. However, I have several concerns about the data that support the authors' conclusions.

Figure 4A: Using bioinformatics analysis, the authors showed that HIF1A, COPS5, and RPS3 are downregulated in MPI cells compared to seven other bat cells and mouse cells. However, the data don't show how much RNA expression of these three genes differs between each bat and mouse cell line. Understanding the expression of these genes in MPI cells compared to each bat species and to the mouse cell line is essential, as the authors propose that a reduction in these genes is crucial for tumor suppression in MPI cells. There may be other bat species in which these genes are significantly downregulated, but we can't see it as the authors did not present expression data. Given that the authors already have RNAseq data from eight bat species and mouse cells, it would be straightforward to depict the expression of these genes in the eight bat species and mouse cells using a bar graph.

Figure 4D: A cell proliferation assay is not the appropriate method to demonstrate the rescue phenotype for the observation in Fig 1B. If a reduction of HIF1A, COPS5, and RPS3 expression in MPI cells is crucial for preventing the transformation seen in Fig 1B - a main focus of this paper - then overexpression of these genes should promote transformation in MPI cells. The authors should overexpress these genes in MPI cells to monitor soft agar colony formation and tumor formation as shown in Fig 1B and Fig 2. If the authors cannot observe enhanced transformation by individual genes, they should consider a combined expression of these genes in MPI cells.

Figure 6A: The control band of COPS5 in MPI cells shows a higher molecular weight compared to that with RAS and LT. The authors suggest that this might be due to splicing variants or modifications. However, it's also possible that this is just a nonspecific band. If this is a nonspecific band, RAS and LT still appear to induce COPS5 well in MPI cells. This is crucial, as the authors' conclusion is that HIF1A can't induce COPS5 due to the lack of a HIF1A binding site near the COPS5 gene. The identity of this band could be easily verified by performing a COPS5 knockout or knockdown in MPI cells to determine whether these high molecular weight bands disappear or not.

Figure 6A: The expression of RPS3 was markedly lower in MPI cells than mouse cells. The authors didn't focus on this very low basal expression of RPS3 in MPI cells because it wasn't increased by RAS in mouse cells and MPI cells. However, it's possible that high expression of RPS3 in mouse cells may cooperate with RAS + LT to enhance transformation in mouse cells but not in MPI cells with very low expression of RPS3. If overexpression of RPS3 in MPI cells promotes transformation in colony formation assay and tumor formation in mice by the introduction of RAS + LT, as I suggested above, the authors may need to consider this possibility.

Fig 6C: The authors suggest that RSC has a HIF1A binding site that contributes to COPS5 induction. However, I don't observe the "CACGT" motif in any of the species displayed, which leads me to question whether this site is indeed a HIF1A binding site. The authors should perform a transactivation assay by HIF1A overexpression as well as a ChIP assay to demonstrate that endogenous HIF1A can bind to this sequence.

Minor comments:

Fig 5B: There are many peaks close to the HIF1A gene that are lost in MPI compared to MSF. I'm unclear why the authors chose only the distant element CNE143336 from the HIF1A gene (190 kb downstream of HIF1A gene). Although a reporter assay showed this element's activity is reduced in MPI cells compared to other cell types, there is no evidence that this element affects HIF1A expression unless the authors knockout this element in cells. The authors might consider removing this data to make the manuscript more focused.

Figure 6B: Why don't the protein levels of HIF1A and COPS5 show significant differences, even though these genes were identified as highly downregulated in MPI cells in Fig 4A?

New figures should not be cited in the discussion section.

Despite COPS5 being the focus of the study, there is no discussion on the COPS5 protein function. As a subunit of the COP9 signalosome, it can control the cellular ubiquitination status. The authors should discuss what is known and unknown about the COP9 signalosome and cancer. Based on Fig. 6A, could it be possible that COPS5 augments the expression of the HIF1A protein rather than HIF1A promoting COPS5? This possibility is based on the COP9 signalosome function that regulates protein degradation.

Fig S7: Although it is not the focus of this study, the strong induction of senescence, p53, and p21 by DNA damage in MPI cells compared to mouse cells is interesting.

Reviewer #3 (Remarks to the Author): Expert in bat genomics and evolution, comparative genomics, genome assembly, and long-read sequencing

In the manuscript Hua et al, the authors investigated cancer resistance in fibroblast cell lines across seven bat species with mouse as the control and ascertained the potential molecular mechanism. By activating oncogenic HRAS and SV40 LT, the authors found that one of seven bat species, *Myotis pilosus*, exhibits cancer resistance. The authors further performed transcriptome analyses and identified a few down-regulated genes in *M. pilosus* which have been demonstrated to inhibit tumour cell growth when silenced. Further sequence, mRNA and protein expression analyses suggest the loss of enhancer upstream of *COPS5* in *M. pilosus* may contribute to their cancer resistance compared to other mammals investigated. In general, the study and experiments were well designed with logics, driven by their sequential results. The conclusions were made based on multiple layers of evidence and thus are robust. However, I have a few suggestions and questions that would like to be clarified by the authors.

The authors didn't mention the rationale of species selection in the manuscript (I presume the selection was based on sample availability?). Based on the topic (cancer resistance /longevity), long-lived and short-lived species are preferably selected for comparison. As far as I am aware only two of the seven species have longevity data recorded (*R. ferrumequinum* and *R. leschenaultii*). Are there any ecological data (including Maximum lifespan etc) available for the remaining bat species, *M. pilosus* in particular? The authors mentioned that *Myotis* bats are generally long-lived, but exceptions exist (e.g. *Myotis nigricans* and *Myotis vivesi* are not long-lived). This information is important to draw accurate conclusions from the analyses and experiments.

Followed by the point above, *R. ferrumequinum* is also a very long-lived species (could possibly live longer than *M. pilosus*) and show low cancer incidence in the field. However, the authors' oncogenic transformation experiments showed *R. ferrumequinum* formed as large colonies as mouse and other species including *R. leschenaultii*, which is not a long-lived species. I am not an expert on cancer experimental biology. But I wonder if cancer can be activated by different approaches (e.g. Ras family mutation or overexpression, TP53 mutation)? If yes, does it indicate different long-lived species can resist cancer with specific molecular mechanisms of cancer activation, but not the other? Without testing these hypothesis, it is less convincing to state " MPI cells are resistant to malignant transformation in contrast to mouse and other six bat species". This point needs to be discussed and the statement needs rephrasing, and the independent evolution of cancer resistance needs to be highlighted.

For WGCNA analyses, the power was weak by using only 24 samples so that the gene co-expression module might not be accurate. I recommend using other methods (e.g. DEG analyses) to confirm the candidate genes (focused below) that showed differentially expressed in *M. pilosus*. This recommendation is based on the fact that the authors initially found 5 hub genes (downregulation in *M. pilosus*) but dropped EP300 and EIF5B later because their downregulation had no effects on cell growth. Actually, these two genes exhibited up-regulation during aging in another extremely long-lived bats (*M.*

myotis) and may be longevity-associated. Please see Supp data one in the paper:
<https://www.nature.com/articles/s41559-019-0913-3>.

The authors investigated the regulatory regions (CNEs) of candidate genes but did not analyse the protein-coding regions of these genes, e.g. any particular regions or site under natural selection? If none of these coding sequences or regions are under selection, down-regulation of these genes was likely caused by loss of enhancer upstream of COPS5 in *M. pilosus*. Looking at protein-coding regions of these genes would strengthen the conclusions drawn by investigating CNEs.

Two minor points:

The authors mainly employed student t tests for statistical analyses throughout. The prerequisite to use t test is that observations should follow normal distribution. Did authors test data normality before applying t test? If non-normal distributed, the authors may think about non-parametric tests such as Mann-Whitney U tests.

A typo under the section 'Downregulated expression of COPS5 reduces tumor size'? The third line: 'uncharged' or 'unchanged'?

Response to the reviewers

We are grateful to the reviewers for their constructive comments, which have helped improve our manuscript. Below please find our point-to-point response (in blue).

Reviewer #1

The authors have investigated cancer resistance in seven bat species by activating oncogenic Ras and SV40 LT in their fibroblasts, and found the bat species *Myotis pilosus* (MPI) is the most resistant. Via comparative transcriptome, they have discovered that three genes HIF1A, COPS5, and RPS3 are downregulated in MPI. They further report that a potential enhancer containing the HIF1A binding site upstream of COPS5 is lost in MPI, and the inhibition of COPS5 expression reduces the xenograft tumor formation in mice. Their work sheds light on the molecular mechanisms of cancer resistance mechanisms in bats, some of which are long-lived species. The manuscript is well-written.

Response:

We thank the reviewer for the positive evaluation.

The following issues need to be addressed:

Comment 1

Does MPI has the longest life span among the 7 species compared?

Response:

This is an interesting question. Unfortunately, we were unable to find any records of the lifespan of MPI from literature and the related databases. Out of the seven bat species, only two have recorded lifespan data: *Rhinolophus ferrumequinum* with a lifespan of 30.5 years and *Rousettus leschenaultii* with a lifespan of 14 years in the AnAge database (genomics.senescence.info/species/index.html). It is worth noting that cells derived from long-lived species have a higher tendency to undergo senescence compared to cells derived from relatively short-lived species (PMID: 31098949). Thus, we induced senescence in the primary cell lines of the seven bat species, as well as mice, using etoposide. We found that the β -galactosidase activity, a commonly used indicator of senescence, was significantly higher in MPI fibroblasts compared to the other six bat species and mouse ($P < 0.01$; Student's t tests; **Supplementary Fig. 8**), which suggests that MPI may have a longer lifespan than the bat species examined. We have included these results in the main text (**page 12, paragraph 1**).

“Unfortunately, there is no record of the lifespan of MPI. However, it is worth noting that cells from long-lived species tend to undergo senescence more than cells from short-lived species³⁵. We thus induced senescence in the primary cell lines of the eight species mentioned above using etoposide. When we examined the commonly used senescence indicator, we found that the β -galactosidase activity was significantly higher in MPI

*fibroblasts compared to the other seven species ($P < 0.01$; Student's t -tests; **Supplementary Fig. 8a, b**). Moreover, the protein levels of p21 and p53, which are the marker genes for cellular senescence^{38,39}, were significantly higher in MPI fibroblasts than in mouse fibroblasts ($P < 0.01$; two-tailed Student's t -tests; **Supplementary Fig. 8c, d**). These results suggest that MPI may have a longer lifespan than the other six bat species examined, as well as mice.”*

Comment 2

What is the rationale using fibroblasts to do the experiment? Most cancers in humans arise from epithelial cells (adenomas or carcinomas), and fibroblasts are among the cell types that consist of tumor microenvironment. How the molecular changes in fibroblasts influence tumor cells should be discussed. Will using fibroblast compromise the interpretation of the survival results in Figure 4B?

Response:

We completely agree that most cancers in humans originate from epithelial cells, and fibroblasts are one of the cell types present in the tumor microenvironment. However, studying fibroblasts can provide insights into whether a species has a tendency to be resistant to cancer by assessing their resistance to malignant transformation. This has been observed in fibroblasts derived from naked mole-rats and blind mole-rats, which are well-known for their anti-cancer properties (PMID: 23129611, PMID: 19858485, and PMID: 23937926). Furthermore, the molecular changes that occur in fibroblasts may play crucial roles in their resistance to malignant transformation in anti-cancer species compared to species prone to cancer (PMID: 23129611, PMID: 19858485, and PMID: 23937926). Indeed, when we inhibited the expression of *COPS5*, which was found to be downregulated in MPI fibroblasts, several human epithelial and tumor cell lines exhibited significantly decreased proliferation (please see the response to the following comment). The survival results of cancer patients presented in Figure 4B were considered as one of the factors for selecting candidate genes that potentially contribute to the resistance of MPI fibroblasts against malignant transformation. We have included this information in the main text to clarify the rationale behind using fibroblasts for our experiments. Additionally, we have discussed the potential impacts the molecular changes identified in MPI fibroblasts on tumor cells.

*“Although most cancers in humans originate from epithelial cells and fibroblasts are part of the tumor microenvironment, studying fibroblasts can provide insights into whether a species tends to have anti-cancer properties by assessing their resistance to malignant transformation. For example, studies have shown that the expression of oncogenic *HRAS(G12V)* and *SV40 large antigen (SV40 LT)* effectively induces malignant transformation in mouse fibroblasts^{13,14}. However, this transformation is not observed in fibroblasts derived from naked mole-rats and blind mole-rats, which are well-known for their anti-cancer characteristics^{15,16}. (page 4, paragraph 1)*

*“In the case of MPI, the decreased expression of *HIF1A* and *COPS5* may represent a novel anticancer mechanism. Indeed, *HIF1A* is known to be overexpressed in different cancer progressions and activates the transcription of genes involved in crucial aspects of cancer biology, such as angiogenesis, cell survival and proliferation, glucose*

metabolism, and invasion^{45,46}. Similarly, COPS5 has been observed to be amplified in several carcinomas and plays an important role in tumorigenesis by interacting with various oncogenic genes, including p53 and p27^{47,48}. The downregulation of HIF1A and COPS5 has been shown to inhibit the proliferation and malignant behaviors of cancer cells, suggesting that inhibitors targeting the two proteins may have potential as anticancer therapeutics^{45,46,49,50}.” (page 13, paragraph 1)

Comment 3

It would be informative to inhibit the expression COPS5 in human epithelial cells to see what happened.

Response:

This is an excellent suggestion. Following the suggestion, we inhibited the expression of COPS5 in human proximal tubular epithelial cells (HK-2), a breast cancer cell line derived from mammary epithelium (MCF-7), and a pancreatic cancer cell line derived from the pancreatic duct epithelium (PANC-1). The results showed a significant decrease in proliferation in these epithelial cell lines. We have added these results to the main text (page 13, paragraph 2).

*“Given that most cancers in humans arise from epithelial cells, rather than fibroblasts, to further explore the relationship between the downregulation of COPS5 observed in fibroblasts and tumor cells, we inhibited the expression of COPS5 in human proximal tubular epithelial cells (HK-2), a breast cancer cell line derived from mammary epithelium (MCF-7), and a pancreatic cancer cell line derived from the pancreatic duct epithelium (PANC-1). The results showed that these epithelial and tumor cell lines exhibited a significant decrease in proliferation ($P < 0.01$, two-tailed Student’s t -tests; **Supplementary Fig. 10**), providing further evidence for the important roles of COPS5 downregulation in the resistance of MPI cells to malignant transformation”*

Comment 4

Bats are known for their genomes harboring many active transposons. Is the loss of the potential enhancer related to transposon insertion/deletion? Is the accelerated evolution of 5,938 CNEs in MPI related to transposon activity?

Response:

Following the suggestion, we have detected transposons in the genome of MPI following a previous method (PMID: 37071810) and the curated de novo transposable element (TE) consensus sequence library (PMID: 33436076). Finally, a total of 3,469,596 TEs in the MPI genome were identified (**Supplementary Table 8**). Among the CNEs under the accelerated evolution in MPI, there were 1,182 overlapping with these TEs, which was significantly less than the ratio of the CNEs that were not accelerated evolution in MPI overlapped with TEs ($P < 0.001$; two-tailed χ^2 test), suggesting that the accelerated evolution of CNEs in MPI was unrelated to transposon activity. Nevertheless, we identified 13 and 31 transposons within 5 kb and 10 kb fragments around the loss of the potential enhancer, respectively. Thus, we cannot rule out the possibility that its loss is related to transposon activity. We have included these results in the main text (page 12, paragraph 2).

Comment 5

Figure 4B is hard to understand: more explanation should be provided in the Figure legends.

Response:

We have revised the legend of Figure 4B as follows.

“According to the KMplot database²¹, the number of cancer types was counted if the upregulated expression of a gene within M1 or the downregulated expression of a gene within M2 significantly prolonged the survival time of cancer patients. The frequency distribution of the ratios of the number of these tumor types to the total number of tumor types with significant survival rates across 350 genes involved in co-expression modules M1 and M2 is shown. The lower expression of the top 5 hub genes, including HIF1A, EP300, EIF5B, COPS5, and RPS3, were generally associated with a higher proportion of survivors with tumors.”

Comment 6

Genes should be in italics.

Response:

Corrected.

Reviewer #2

This is an interesting study aimed at identifying bat species that are resistant to cancer development and exploring the underlying mechanisms of their resistance. The authors demonstrated that out of eight bat species, *Myotis pilosus* (MPI) cells exhibited particularly strong resistance to cancer. In comparison to other species and mouse cells, MPI cells demonstrated a strong resistance to transformation caused by oncogenic Ras and SV40 LT. Through a combination of bioinformatics and transcriptomics analyses, they revealed that HIF1A, COPS5, and RPS3 are downregulated in MPI cells and that depletion of these genes in mouse cells reduces proliferation. Additionally, the study showed that the HIF1A binding motif located upstream of the COPS5 gene is lost in MPI cells. They interpreted that MPI cells fail the HIF1A-mediated COPS5 induction after the introduction of Ras and SV40 LT, which may prevent cancer progression. In line with this, they demonstrated that knockout COPS5 in mouse cell lines reduced their tumor formation in mice. Some mammals have unique ways of preventing tumors. This is a growing area of interest, as we may be able to learn from these strategies to develop new cancer preventions/treatments for humans. Therefore, their findings may be very interesting in this field. However, I have several concerns about the data that support the authors' conclusions.

Response:

We thank the reviewer for the positive evaluation.

Comment 1

Figure 4A: Using bioinformatics analysis, the authors showed that HIF1A, COPS5, and RPS3 are downregulated in MPI cells compared to seven other bat cells and mouse cells. However, the data don't show how much RNA expression of these three genes differs between each bat and mouse cell line. Understanding the expression of these genes in MPI cells compared to each bat species and to the mouse cell line is essential, as the authors propose that a reduction in these genes is crucial for tumor suppression in MPI cells. There may be other bat species in which these genes are significantly downregulated, but we can't see it as the authors did not present expression data. Given that the authors already have RNAseq data from eight bat species and mouse cells, it would be straightforward to depict the expression of these genes in the eight bat species and mouse cells using a bar graph.

Response:

This is an excellent suggestion. We have included a bar graph (**Supplementary Fig. 4**) in the main text, which presents the expression data of the top 5 hub genes, including *HIF1A*, *COPS5*, and *RPS3*. The graph clearly showed that these genes have significantly lower expression levels in MPI cells compared to the cells of other bat species.

Comment 2

Figure 4D: A cell proliferation assay is not the appropriate method to demonstrate the rescue phenotype for the observation in Fig 1B. If a reduction of HIF1A, COPS5, and RPS3 expression in MPI cells is crucial for preventing the transformation seen in Fig 1B - a main focus of this paper - then overexpression of these genes should promote

transformation in MPI cells. The authors should overexpress these genes in MPI cells to monitor soft agar colony formation and tumor formation as shown in Fig 1B and Fig 2. If the authors cannot observe enhanced transformation by individual genes, they should consider a combined expression of these genes in MPI cells.

Response:

We agree with the reviewer that the cell proliferation assay is not the exact method to demonstrate the rescue phenotype for the observation in Fig 1B. We have removed the term “rescue” and revised the sentence as follows:

“To further investigate the involvement of HIF1A, COPS5, RPS3 in the resistance of MPI cells to malignant transformation, we overexpressed HIF1A, COPS5, and RPS3 in MPI-SF^{HRAS\SV40LT} ...”

Following the suggestion, we overexpressed *HIF1A*, *COPS5*, and *RPS3* in MPI cells to assess their impact on soft agar colony formation and tumor formation. Although there were no remarkable differences observed with the overexpression of individual *HIF1A*, *COPS5*, and *RPS3*, the combined overexpression of the three genes in MPI cells resulted in significantly larger colonies *in vitro* ($P < 0.001$; two-tailed Student’s *t*-tests;

Supplementary Fig. 6b, and c) and larger tumor size *in vivo* ($P < 0.05$; two-tailed Student’s *t*-tests; **Supplementary Fig. 6d, e, f, and g**) compared to the control groups. These results further support the involvement of *HIF1A*, *COPS5*, *RPS3* in the resistance of MPI cells to malignant transformation. We have added these results to the main text (**page 7, paragraph 1**).

Comment 3

Figure 6A: The control band of COPS5 in MPI cells shows a higher molecular weight compared to that with RAS and LT. The authors suggest that this might be due to splicing variants or modifications. However, it's also possible that this is just a nonspecific band. If this is a nonspecific band, RAS and LT still appear to induce COPS5 well in MPI cells. This is crucial, as the authors' conclusion is that HIF1A can't induce COPS5 due to the lack of a HIF1A binding site near the COPS5 gene. The identity of this band could be easily verified by performing a COPS5 knockout or knockdown in MPI cells to determine whether these high molecular weight bands disappear or not.

Response:

This is a great suggestion. Following this suggestion, we inhibited the expression of *COPS5* in MPI cells and found that the intensity of this band was significantly weakened ($P < 0.01$, two-tailed Student’s *t*-tests; **Supplementary Fig. 11**). This result suggests that the observed difference in protein size did not result from the non-specificity of the COPS5 antibody. We have added the result to the main text (**page 14, paragraph 1**).

Comment 4

Figure 6A: The expression of RPS3 was markedly lower in MPI cells than mouse cells. The authors didn't focus on this very low basal expression of RPS3 in MPI cells because it wasn't increased by RAS in mouse cells and MPI cells. However, it's possible that high expression of RPS3 in mouse cells may cooperate with RAS + LT to enhance

transformation in mouse cells but not in MPI cells with very low expression of RPS3. If overexpression of RPS3 in MPI cells promotes transformation in colony formation assay and tumor formation in mice by the introduction of RAS + LT, as I suggested above, the authors may need to consider this possibility.

Response:

We agree the reviewer that we cannot completely disregard the potential roles of RPS3 in the resistance of MPI cells to malignant transformation. In particular, the overexpression of *RPS3* resulted in a significant increase in cell proliferation (**Fig. 4d**). Furthermore, the combined overexpression of *HIF1A*, *COPS5*, and *RPS3* in MPI cells led to the formation of significantly larger soft agar colonies and tumors compared to the control groups (**Supplementary Fig. 6**). We have included a discussion of the potential roles of *RPS3* in the resistance of MPI cells to malignant transformation (**page 13, paragraph 1**).

Comment 5

Fig 6C: The authors suggest that RSC has a HIF1A binding site that contributes to COPS5 induction. However, I don't observe the "CACGT" motif in any of the species displayed, which leads me to question whether this site is indeed a HIF1A binding site. The authors should perform a transactivation assay by HIF1A overexpression as well as a ChIP assay to demonstrate that endogenous HIF1A can bind to this sequence.

Response:

The Homer tool (PMID: 20513432) predicted the binding motif of HIF1A in the RSC sequences. Although the motif scores varied among different species, there was statistical significance in the possibility of binding affinity with HIF1A. We have added the motif scores and the corresponding statistical results to the revised **Figure 6c**.

We followed the suggestion to confirm the binding ability of this motif with HIF1A. Firstly, we generated a stable cell line of HEK293T that continuously expresses HIF1A. Using this cell model, we conducted a dual-luciferase assay to assess the regulatory activity of RSC with or without this binding motif. When we deleted the 17 bp fragment containing this binding motif, the regulatory activity of mouse RSC significantly decreased ($P < 0.001$; Student's *t*-test; **Fig. 6f**). Further, when we respectively added the 17 bp fragments of mouse and another bat (RSI) to the RSC of MPI, the regulatory activity of the edited RSCs of MPI significantly enhanced ($P < 0.05$; Student's *t*-tests; **Fig. 6f**). Additionally, we performed a ChIP-qPCR to demonstrate the binding ability of endogenous HIF1A to this binding motif. We used an antibody specific to HIF1A to enrich HIF1A along with its DNA targets in mouse and MPI fibroblasts, respectively. After purification, we used specific primers designed based on the mouse and MPI RSC sequences around the 17 bp deletion to perform qPCR. Upon comparison, we found that the concentration of this fragment of mouse RSC was significantly higher than that of MPI RSC ($P < 0.001$, two-tailed Student's *t*-test; **Fig. 6g, h**). Overall, these results strongly support the presence of the binding site of HIF1A in the examined fragment of mice RSC but its absence in that of MPI RSC sequence. We have included these results in the main text (**page 10, paragraph 1**).

Comment 6

Fig 5B: There are many peaks close to the HIF1A gene that are lost in MPI compared to MSF. I'm unclear why the authors chose only the distant element CNE143336 from the HIF1A gene (190 kb downstream of HIF1A gene). Although a reporter assay showed this element's activity is reduced in MPI cells compared to other cell types, there is no evidence that this element affects HIF1A expression unless the authors knockout this element in cells. The authors might consider removing this data to make the manuscript more focused.

Response:

We focused on this peak because it was the only one that overlapped with the evolutionarily conserved non-coding elements (CNEs) that experienced accelerated evolution in MPI, namely CNE143336. We have revised the relevant sentences to clarify this point.

Following the suggestion, we designed guide RNAs based on the sequence of CNE143336 to perform the CRISPR-dCas9 technology to block its regulatory activity. The results showed that when the regulatory activity of CNE143336 was blocked, the expression level of *HIF1A* significantly decreased ($P < 0.001$, two-tailed Student's *t*-test; **Fig. 5d, e, and f**). These findings further confirm that CNE143336 indeed affects the expression of HIF1A. We have included these results in the main text (**page 8, paragraph 2**).

Comment 7

Figure 6B: Why don't the protein levels of HIF1A and COPS5 show significant differences, even though these genes were identified as highly downregulated in MPI cells in Fig 4A?

Response:

The protein levels of HIF1A and COPS5 did show significant differences between mouse and MPI fibroblasts. However, due to the relatively long exposure time, these differences were reduced. We have updated a new Western blot image and marked their statistical differences in the revised **Figure 6B**.

Comment 8

New figures should not be cited in the discussion section.

Response:

The experimental results regarding the induction of cell senescence, while not the primary focus of our study, provide valuable insights into the potential evolutionary mechanisms behind MPI's anti-cancer properties. Therefore, we have included these results in the discussion section as a supplementary figure.

Comment 9

Despite COPS5 being the focus of the study, there is no discussion on the COPS5 protein function. As a subunit of the COP9 signalosome, it can control the cellular ubiquitination status. The authors should discuss what is known and unknown about the COP9 signalosome and cancer. Based on Fig. 6A, could it be possible that COPS5 augments the

expression of the HIF1A protein rather than HIF1A promoting COPS5? This possibility is based on the COP9 signalosome function that regulates protein degradation.

Response:

This is an excellent suggestion. Indeed, we observed that when inhibiting the expression of *COPS5*, there was a significant decrease in *HIF1A* expression (**Supplementary Fig. 9**). This result suggests a more complex relationship between *COPS5* and *HIF1A* than previously understood. We have included this result in the main text, along with a discussion on the function of *COPS5* and the possibility of *COPS5* enhancing the expression of *HIF1A* protein (**page 13, paragraph 1**).

*“COPS5 has been observed to be amplified in several carcinomas and plays an important role in tumorigenesis by interacting with various oncogenic genes, including p53 and p27^{47,48}. The downregulation of HIF1A and COPS5 has been shown to inhibit the proliferation and malignant behaviors of cancer cells, suggesting that inhibitors targeting the two proteins may have potential as anticancer therapeutics^{45,46,49,50}. Our findings suggest that HIF1A may promote the expression of COPS5. However, it is also possible that COPS5 may enhance the stability and abundance of HIF1A, as it is known to stabilize HIF1A²⁷. Indeed, when inhibiting the expression of COPS5, we observed a significant decrease in HIF1A expression ($P < 0.01$, two-tailed Student’s t-test; **Supplementary Fig. 9**).”*

Comment 10

Fig S7: Although it is not the focus of this study, the strong induction of senescence, p53, and p21 by DNA damage in MPI cells compared to mouse cells is interesting.

Response:

We are glad that the reviewer considers this result of our study to be interesting.

Reviewer #3

In the manuscript Hua et al, the authors investigated cancer resistance in fibroblast cell lines across seven bat species with mouse as the control and ascertained the potential molecular mechanism. By activating oncogenic HRAS and SV40 LT, the authors found that one of seven bat species, *Myotis pilosus*, exhibits cancer resistance. The authors further performed transcriptome analyses and identified a few down-regulated genes in *M. pilosus* which have been demonstrated to inhibit tumour cell growth when silenced. Further sequence, mRNA and protein expression analyses suggest the loss of enhancer upstream of COPS5 in *M. pilosus* may contribute to their cancer resistance compared to other mammals investigated. In general, the study and experiments were well designed with logics, driven by their sequential results. The conclusions were made based on multiple layers of evidence and thus are robust. However, I have a few suggestions and questions that would like to be clarified by the authors.

Response:

We thank the reviewer for the positive evaluation.

Comment 1

The authors didn't mention the rationale of species selection in the manuscript (I presume the selection was based on sample availability?). Based on the topic (cancer resistance /longevity), long-lived and short-lived species are preferably selected for comparison. As far as I am aware only two of the seven species have longevity data recorded (*R. ferrumequinum* and *R. leschenaultii*). Are there any ecological data (including Maximum lifespan etc) available for the remaining bat species, *M. pilosus* in particular? The authors mentioned that *Myotis* bats are generally long-lived, but exceptions exist (e.g. *Myotis nigricans* and *Myotis vivesi* are not long-lived). This information is important to draw accurate conclusions from the analyses and experiments.

Response:

It is indeed true that the seven species examined in this study are common bats in the southwest region of China, where our institute is located. We have included information about the availability of samples in the Methods section.

We fully acknowledge the importance of having information about the lifespan of *M. pilosus* for our study. However, despite our thorough search of literature and databases, we were unable to find ecological data on the lifespan of *M. pilosus*. Notably, cells derived from long-lived species have a higher tendency to undergo senescence compared to cells derived from relatively short-lived species (PMID: 31098949). Thus, we induced senescence in the primary cell lines of the seven bat species, as well as mice, using etoposide. We found that the β -galactosidase activity, a commonly used indicator of senescence, was significantly higher in MPI fibroblasts compared to the other six bat species and mouse ($P < 0.01$; Student's *t* tests; **Supplementary Fig. 8a, b**). Additionally, the protein levels of p21 and p53, which are marker genes for cellular senescence, were significantly higher in MPI fibroblasts than in mouse fibroblasts ($P < 0.01$; two-tailed Student's *t*-tests; **Supplementary Fig. 8c, d**). These findings suggest that MPI may have

a longer lifespan than the other six bat species examined, as well as mice. We have included these results in the main text (page 12, paragraph 1).

Comment 2

Followed by the point above, *R. ferrumequinum* is also a very long-lived species (could possibly live longer than *M. pilosus*) and show low cancer incidence in the field. However, the authors' oncogenic transformation experiments showed *R. ferrumequinum* formed as large colonies as mouse and other species including *R. leschenaultii*, which is not a long-lived species. I am not an expert on cancer experimental biology. But I wonder if cancer can be activated by different approaches (e.g. Ras family mutation or overexpression, TP53 mutation)? If yes, does it indicate different long-lived species can resist cancer with specific molecular mechanisms of cancer activation, but not the other? Without testing these hypothesis, it is less convincing to state "MPI cells are resistant to malignant transformation in contrast to mouse and other six bat species". This point needs to be discussed and the statement needs rephrasing, and the independent evolution of cancer resistance needs to be highlighted.

Response:

This is an excellent suggestion. The overexpression of the oncogenic HRAS(G12V) and SV40 large antigen (SV40 LT) is indeed a commonly-used approach to induce malignant transformation of cells, especially in non-model organisms like naked mole-rats and blind mole-rats (PMID: 19858485, PMID: 23937926). We agree with the reviewer that other RAS mutations and oncogenic genes may induce malignant transformation in species-specific ways. Following the suggestion, we have rephrased the relevant sentences and included a discussion on this point in the main text (page 11, paragraph 2).

"It is important to note that while the oncogenic HRAS(G12V) is the most common cancer-associated substitution and is also a commonly used method to induce malignant transformation of cells³², particularly in non-model organisms like naked mole-rats and blind mole-rats^{12,13}, neoplastic transformation of cells may require species- or cell type-specific mutations of HRAS or other RAS members^{10,32}. Therefore, our results do not completely exclude the possibility that the other six bat species examined in this study may have evolved different anti-cancer mechanisms compared to MPI. Overall, our findings not only provide direct support for the hypothesis that bats have evolved cancer resistance^{5,6,8,20}, but also suggest that this resistance may have independently evolved in various bat lineages^{33,34}."

Comment 3

For WGCNA analyses, the power was weak by using only 24 samples so that the gene co-expression module might not be accurate. I recommend using other methods (e.g. DEG analyses) to confirm the candidate genes (focused below) that showed differentially expressed in *M. pilosus*. This recommendation is based on the fact that the authors initially found 5 hub genes (downregulation in *M. pilosus*) but dropped EP300 and EIF5B later because their downregulation had no effects on cell growth. Actually, these two genes exhibited up-regulation during aging in another extremely long-lived bats (*M. myotis*) and may be longevity-associated. Please see Supp data one in the paper: <https://www.nature.com/articles/s41559-019-0913-3>.

Response:

Following the suggestion, we performed DEG analyses between these species and observed that 93.8% of genes in M1, which had the highest module eigengene (ME) value, were significantly up-regulated in *M. pilosus*. And 42.9% of genes in M2, which had the lowest ME value, were significantly down-regulated in *M. pilosus* (**Supplementary Table 1**). The results were largely consistent with the co-expression trends reflected through the WGCNA analysis.. Furthermore, the expression levels of the five candidate genes from M2 were significantly lower in *M. pilosus* compared to the other six bat species (**Supplementary Fig. 4**). We have added these results to the main text (**page 6, paragraph 1**).

Additionally, we discussed the up-regulation of *EP300* and *EIF5B* during aging in the long-lived bat (*M. myotis*) and cited this paper.

Comment 4

The authors investigated the regulatory regions (CNEs) of candidate genes but did not analyse the protein-coding regions of these genes, e.g. any particular regions or site under natural selection? If none of these coding sequences or regions are under selection, down-regulation of these genes was likely caused by loss of enhancer upstream of COSP5 in *M. pilosus*. Looking at protein-coding regions of these genes would strengthen the conclusions drawn by investigating CNEs.

Response:

Following the suggestion, we utilized the PAML software (PMID: 17483113) to assess evolutionary pressures on the protein-coding regions of the five candidate genes. Our analysis revealed that none of these genes were under positive selection or the accelerated evolution (**Supplementary Table 4**), providing further support for the notion that the down-regulation of these genes may play important roles in cellular transformation, rather than being driven by selection pressure on their protein-coding sequences. We have included these results in the main text (**page 6, paragraph 2**).

“To provide additional evidence supporting the potential influence of gene expression regulation, rather than sequence alterations, on cancer progression, we detected evolutionary pressures on the protein-coding regions of the five candidate genes using PAML¹⁹. Our analyses revealed that none of these genes were under positive selection or exhibited accelerated evolution in MPI (Supplementary Table 4)”

Two minor points:

Comment 5

The authors mainly employed student t tests for statistically analyses throughout. The prerequisite to use t test is that observations should follow normal distribution. Did authors test data normality before applying t test? If non-normal distributed, the authors may think about non-parametric tests such as Mann-Whitney U tests.

Response:

Because we conducted three or more biological replicates for each experiment, it is appropriate to use the Student's *t*-test for statistical analyses. So, there is no concern here.

Comment 6

A typo under the section 'Downregulated expression of COPS5 reduces tumor size'? The third line: 'uncharged' or 'unchanged'?

Response:

Corrected.

REVIEWERS' COMMENTS

Reviewer #1 (Remarks to the Author):

I thank the authors for effectively addressing all of my concerns.

Reviewer #2 (Remarks to the Author):

The authors have faithfully and adequately addressed all my concerns raised for this manuscript. These include the gene expression of HIF1A, COPS5, and RPS3 in other bat species, transformation assays both in vitro and in vivo, validation of the COPS5 antibody, transactivation assays by HIF1A overexpression, CHIP assays, and gene knockout of the CNE143336 element, as well as HIF1A regulation by COPS5. It was surprising to me that the distant element CNE143336, located 190 kb downstream of the HIF1A gene, affects the expression of HIF1A. However, the data are quite solid, demonstrating this through the knockout of CNE143336. Since HIF1A deregulation is critically important for the development of a subset of human cancers, it is intriguing to explore the underlying mechanisms for future research.

I do not have any further concerns regarding this manuscript, and I believe it is ready for publication.

Reviewer #3 (Remarks to the Author):

The authors have addressed my concerns. I congratulate the authors on conducting this nice piece of work!

Responses to the reviewers

Reviewer #1:

Comment

I thank the authors for effectively addressing all of my concerns.

Response:

We thank the reviewer for reviewing our revised manuscript.

Reviewer #2:

Comment

The authors have faithfully and adequately addressed all my concerns raised for this manuscript. These include the gene expression of HIF1A, COPS5, and RPS3 in other bat species, transformation assays both in vitro and in vivo, validation of the COPS5 antibody, transactivation assays by HIF1A overexpression, ChIP assays, and gene knockout of the CNE143336 element, as well as HIF1A regulation by COPS5. It was surprising to me that the distant element CNE143336, located 190 kb downstream of the HIF1A gene, affects the expression of HIF1A. However, the data are quite solid, demonstrating this through the knockout of CNE143336. Since HIF1A deregulation is critically important for the development of a subset of human cancers, it is intriguing to explore the underlying mechanisms for future research.

I do not have any further concerns regarding this manuscript, and I believe it is ready for publication.

Response:

We thank the reviewer for reviewing our revised manuscript and for the recommendation of publication.

Reviewer #3:

The authors have addressed my concerns. I congratulate the authors on conducting this nice piece of work!

Response:

We thank the reviewer for reviewing our revised manuscript.